# Src inhibition attenuates polyglutamine-mediated neuromuscular degeneration in spinal and bulbar muscular atrophy

Madoka Iida[1,2], Kentaro Sahashi[1], Naohide Kondo[1], Hideaki Nakatsuji[1], Genki Tohnai[1], Yutaka Tsutsumi[1], Seiya Noda[1,3], Ayuka Murakami[1,3], Kazunari Onodera[1,4], Yohei Okada[1,4,5], Masahiro Nakatochi[6], Yuka Tsukagoshi Okabe[7], Shinobu Shimizu[7], Masaaki Mizuno[7], Hiroaki Adachi[8], Hideyuki Okano[5], Gen Sobue[9] & Masahisa Katsuno[1]

Spinal and bulbar muscular atrophy (SBMA) is a neuromuscular disease caused by an expanded CAG repeat in the androgen receptor (AR) gene. Here, we perform a comprehensive analysis of signaling pathways in a mouse model of SBMA (AR-97Q mice) utilizing a phosphoprotein assay. We measure the levels of 17 phosphorylated proteins in spinal cord and skeletal muscle of AR-97Q mice at three stages. The level of phosphorylated Src (p-Src) is markedly increased in the spinal cords and skeletal muscles of AR-97Q mice prior to the onset. Intraperitoneal administration of a Src kinase inhibitor improves the behavioral and histopathological phenotypes of the transgenic mice. We identify p130Cas as an effector molecule of Src and show that the phosphorylated p130Cas is elevated in murine and cellular models of SBMA. These results suggest that Src kinase inhibition is a potential therapy for SBMA.

[1] Department of Neurology, Nagoya University Graduate School of Medicine, 65 Tsurumai-cho, Showa-ku, Nagoya city, Aichi 466-8550, Japan. [2] Japan Society for the Promotion of Science, 5-3-1, Kojimachi, Chiyoda-ku, Tokyo 102-0083, Japan. [3] Department of Neurology, National Hospital Organization Suzuka National Hospital, 3-2-1, Kasado, Suzuka city, Mie 513-8501, Japan. [4] Department of Neurology, Aichi Medical University School of Medicine, 1, Karimata, Yazako, Nagakute-city, Aichi 480-1195, Japan. [5] Department of Physiology, Keio University School of Medicine, 35, Shinanomachi, Shinjuku-ku, Tokyo 160-8582, Japan. [6] Department of Nursing, Nagoya University Graduate School of Medicine, 1-1-20 Daiko-Minami, Higashi-ku, Nagoya city, Aichi 461-8673, Japan. [7] Department of Advanced Medicine, Nagoya University Hospital, 65 Tsurumai-cho, Showa-ku, Nagoya city, Aichi 466-8560, Japan. [8] Department of Neurology, University of Occupational and Environmental Health School of Medicine, 1-1, Iseigaoka, Yahatanichi-ku, Kitakyushu-city, Fukuoka 807-0804, Japan. [9] Brain and Mind Research Center, Nagoya University, 65 Tsurumai-cho, Showa-ku, Nagoya city, Aichi 466-8550, Japan. Correspondence and requests for materials should be addressed to M.K. (email: ka2no@med.nagoya-u.ac.jp)

Spinal and bulbar muscular atrophy (SBMA) is an X-linked and adult-onset neuromuscular disease caused by abnormal CAG repeat expansions in the androgen receptor (AR) gene[1–3]. SBMA is characterized by muscle weakness, atrophy, and fasciculation of the limb and bulbar muscles[4].

Ligand-dependent toxicity of the pathogenic polyglutamine-expanded AR protein is central to the pathogenesis of SBMA, and anti-androgen therapies have been tested in both basic and clinical studies[5]. Leuprorelin acetate, a luteinizing hormone-releasing hormone (LHRH) agonist, potentially suppresses neurological symptoms in SBMA patients[6,7]. However, the benefit of this drug is limited by its side effects, which include sexual dysfunction and anti-anabolic actions on skeletal muscle. The degenerative mechanism of SBMA is associated with dysregulated proteolysis and mitochondrial dysfunction. However these insights have yet to be applied in the clinic, partially due to the toxicity and low bioavailability of candidate drugs.

Recent studies have provided strong evidence that neuromuscular degeneration occurs in patients with SBMA. Patients with SBMA exhibit elevated serum creatine kinase levels, and their skeletal muscle biopsies show both neurogenic and myopathic changes[8]. Skeletal muscle pathology occurs prior to neurodegeneration in a knock-in mouse model of SBMA[9]. Muscle-specific overexpression of insulin-like growth factor-1 (IGF-1) suppresses motor neuron degeneration in an SBMA mouse model, whereas overexpression of wild-type rat AR in muscle induces motor neuron damage in mice[10,11]. Moreover, muscle-specific excision of the mutant AR gene improves motor phenotype in a BAC transgenic mouse model of SBMA[12]. Peripheral gene silencing of the mutant AR by antisense oligonucleotides also alleviates neurodegeneration in a knock-in mouse model of SBMA, while gene silencing in the central nervous system also results in attenuation of neuromuscular phenotype of a transgenic mouse harboring polyglutamine-expanded AR[13,14]. In humans, the AR mutant suppresses the transcription of the *SLC6A8* gene encoding a creatine transporter, leading to the inhibition of the muscular uptake of creatine[15]. Based on these observations, both motor neurons and skeletal muscle undergo degeneration in subjects with SBMA. However, the pathological mechanism by which the polyglutamine-expanded AR mutant induces neuromuscular degeneration remains to be elucidated.

Here, we aim to clarify the upstream molecular changes involved in the pathological process of SBMA in neuronal and muscular tissues and to develop therapeutics targeting the cardinal pathogenesis of SBMA. Therefore, we perform a comprehensive analysis of signaling pathways in a mouse model of SBMA. Protein phosphorylation plays a major role in regulating protein functions in both neural and non-neural tissues[16], which led us to investigate the role of kinase activations in the pathogenesis of SBMA. In the assay, we find that Src kinase is a pivotal therapeutic target for neuromuscular degeneration in mice with SBMA. The pathogenic AR mutant activates Src signaling, whereas pharmacological suppression of Src improves the viability of cellular models of SBMA and ameliorates the phenotype of the model mice. Furthermore, the phosphorylation of p130Cas, an effector of Src, plays a critical role in the pathogenesis of SBMA. Src kinase inhibitors (SKI) thus are candidate therapeutics for SBMA.

## Results

**Src is activated in cellular and mouse models of SBMA.** We analyzed a set of 17 phosphorylated proteins that are well known to play essential roles in cellular survival and function. Using Bio-Plex phosphorylation protein arrays, we investigated the alterations in various signaling pathways in the spinal cord and skeletal muscle of AR-97Q mice at three stages: pre-onset (6 weeks of age), early symptomatic (9 weeks of age) and advanced (13 weeks of age). The data were compared with data obtained from wild-type mice. We identified altered phosphorylation of several proteins in the array; however, the most outstanding changes were observed in the Src signaling pathway (Fig. 1a). We detected five molecules that showed significantly upregulated phosphorylation (>1.15-fold) in the spinal cords of AR-97Q mice prior to the onset of the disease compared with that in wild-type mice: phosphorylated Src (p-Src), p-Stat3, p-p38MAPK, p-Akt, and p-IκBα. Of these five phosphorylated proteins, only p-Src displayed sustained, significant activation in the spinal cords of SBMA mice until the advanced stage of the disease. In skeletal muscle, the phosphorylation of four proteins, p-Stat3, p-IRS-1, p-GSK, and p-JNK was significantly upregulated (>1.15-fold) in SBMA mice at the pre-onset stage of the disease compared with that in wild-type mice. Notably, the level of phosphorylated Stat3, which is directly activated by Src, was markedly elevated in both spinal cord and skeletal muscle in AR-97Q mice prior to the onset of neurological symptoms compared with the level in wild-type mice. Levels of p-Src were also elevated in the skeletal muscles of AR-97Q mice before and around the time of disease onset compared with the levels in wild-type mice. We thus considered that upregulation of the Src pathway exerts a strong impact on the pathophysiology of SBMA.

We investigated the spinal cords and skeletal muscles of 6-week-old SBMA transgenic mice and wild-type mice using immunoblotting with antibodies against p-Src, Stat3 and p38MAPK and with antibodies against the total proteins for normalization to confirm the presence of increased levels of phosphorylated proteins (Fig. 1b, c). Levels of p-Src and Stat3 were significantly elevated in the spinal cords of AR-97Q mice compared with wild-type mice, according to the quantitative densitometry analysis (Fig. 1d). Levels of p-Src, but not phosphorylated Stat3, were significantly increased in the skeletal muscle of AR-97Q mice compared with wild-type mice (Fig. 1e). Increased p-Src immunoreactivity was detected in spinal motor neurons and skeletal muscles of the transgenic mice compared with the wild-type mice (Fig. 1f, g). Moreover, the levels of p-Src were increased in autopsy specimens of spinal cord and skeletal muscle from patients with SBMA compared with specimens from control patients (Fig. 1h, i). We further compared the level of p-Src in the biopsied skeletal muscles of patients with SBMA and those of control subjects with immunoblotting. Biopsy was performed within 12 months and 7 years from the onset of weakness in SBMA patients, who were genetically diagnosed as SBMA later on. p-Src was upregulated in skeletal muscles of patients with SBMA compared with those of control subjects, indicating that Src phosphorylation is elevated in the skeletal muscle of SBMA patients at an early stage, in agreement with findings in the mouse model of SBMA (Fig. 1j).

Subsequently, we transfected a neuronal cell line, NSC34, and a muscle cell line, C2C12, with full-length human AR containing 24 or 97 CAG repeats, thereby generating stable cell lines harboring AR-24Q as control cells and cells harboring AR-97Q as a cellular model of SBMA (Fig. 2a). The expression levels of AR were upregulated by dihydrotestosterone (DHT) in 24Q and 97Q cells (Fig. 2b). A dual luciferase reporter assay revealed that DHT treatment upregulates the transcriptional activity of 24Q and 97Q cells (Supplementary Fig. 1). Using western blotting, we analyzed the levels of total and p-Src, Stat3 and p38MAPK in these cellular models after culture in the presence or absence of DHT (Fig. 2c). In the presence of DHT, the ratios of p-Src and Stat3 to the total amounts of these proteins were significantly increased in the cellular models of SBMA compared with control cells (Fig. 2d, e). Quantitative real-time polymerase chain reaction (RT-PCR)

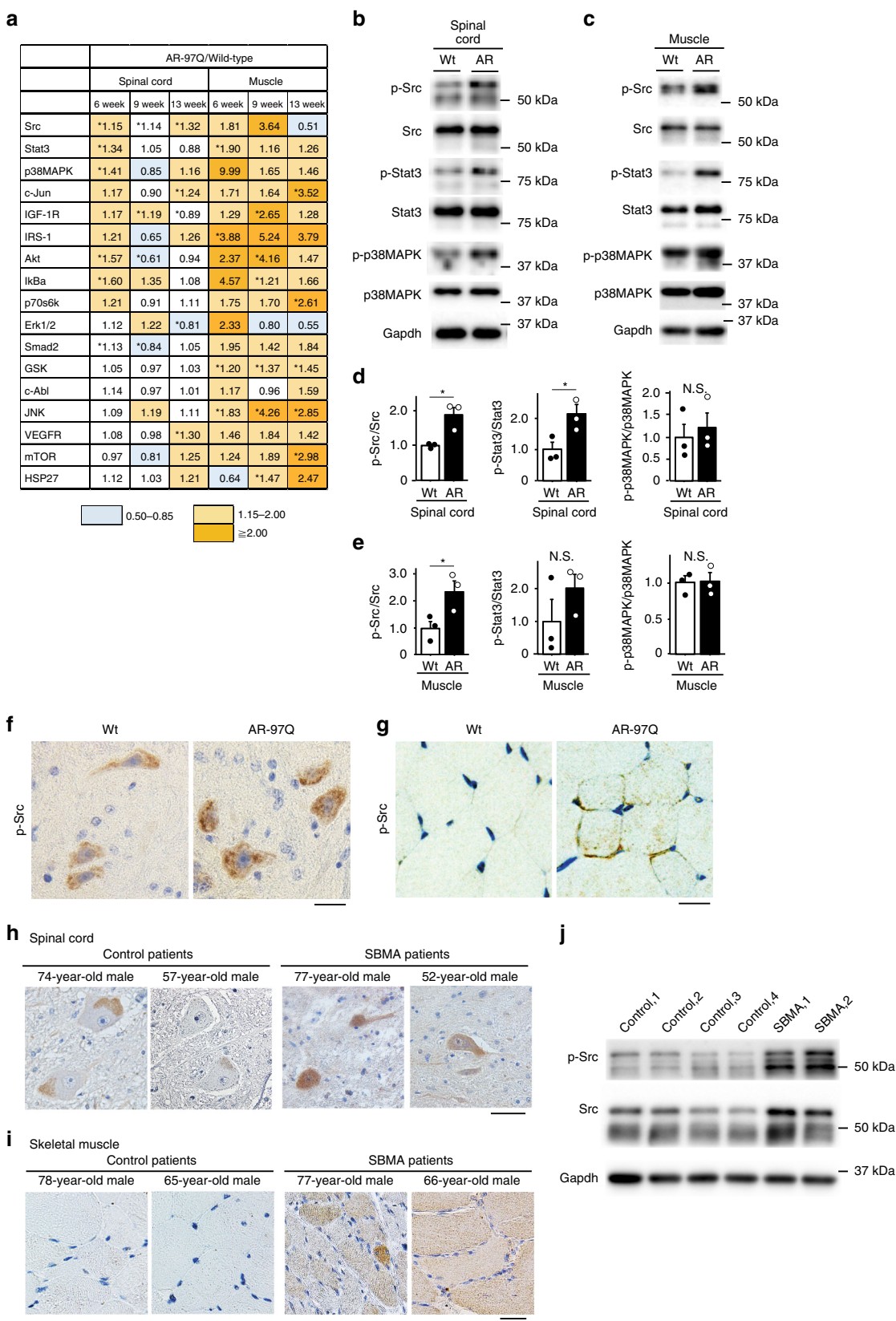

analyses showed that mRNA levels of *Src* in 97Q cells were not significantly different from those in 24Q cells, suggesting that AR regulates Src activity at a post-translational phase (Fig. 2f, g).

Moreover, induced pluripotent stem cell (iPSC)-derived motor neurons from patients with SBMA displayed an increased level of p-Src compared with the level in motor neurons derived from the iPSCs from healthy controls (Fig. 2h, i). The overexpression of AR-97Q in human myogenic cells clone Hu5/KD3 also elevated the level of p-Src compared with the cells overexpressing AR-24Q (Fig. 2j, k). Because the antibody against phospho-Src used in this

**Fig. 1** Abnormally phosphorylated proteins in AR-97Q mice. **a** We analyzed the levels of phosphorylated proteins in the mouse model of SBMA which carries human androgen receptor (AR) with 97 glutamines (AR-97Q) at 6, 9, and 13 weeks of age and compared them to wild-type mice at the same ages using the Bio-Plex phosphoprotein assay ($n = 3$ animals per group). The $p$-values in each of the spinal cord and skeletal muscle were adjusted for false discovery rate (FDR) based on the Benjamini and Hochberg method to correct for multiple testing. *FDR < 0.1. **b**–**e** Immunoblots showing the levels of phosphorylated and total Src, Stat3 and p38MAPK proteins in the spinal cord (**b**) and skeletal muscle (**c**) of wild-type and AR-97Q mice at 6 weeks of age. The quantitative analysis was performed using densitometry ($n = 3$ mice per group) (**d**, **e**). **f**, **g** Immunohistochemistry for p-Src in the spinal cord (**f**) and skeletal muscle (**g**) of 6-week-old mice. **h**, **i** Immunohistochemistry for p-Src in spinal cord (**h**) and skeletal muscle (**i**) samples from control subjects and patients with SBMA. **j** Immunoblots for p-Src and Src in the biopsied muscle specimens of control subjects and patients with SBMA. Error bars indicate the s.e.m. Comparison between AR-97Q and wild-type groups was performed using the unpaired two-sided $t$-test. *$p < 0.05$ (**d**, **e**). SBMA, spinal and bulbar muscular atrophy; N.S., not significant; Wt, wild-type; AR, AR-97Q. Scale bars: 25 μm (**f**, **g**), 40 μm (**h**) and 20 μm (**i**). Source data are provided as a Source Data file (**a**–**e**, **j**)

study may have cross-reacted with other Src family members, including Lyn, Fyn, Lck, Yes, and Hck, we immunoprecipitated cultured cells using a Src-specific antibody and performed western blotting. This result confirmed the selective increase in levels of p-Src, but not other SFKs, in cellular models of SBMA. Immunoblots performed using antibodies against Src family proteins showed increased Src phosphorylation in 97Q cells (Fig. 2l, m).

**An SKI suppresses pathogenic AR toxicity**. To clarify the role of Src activation in the pathophysiology of SBMA, we next examined the effect of an SKI on polyglutamine-mediated cytotoxicity in neuronal and muscle cell models of SBMA. NSC34 and C2C12 cells stably expressing AR-97Q displayed diminished cellular viability, decreased cell number and increased lactate dehydrogenase (LDH) release compared with cells of the same lines that stably expressed AR-24Q (Fig. 3a–f). Treatment with A419259 trihydrochloride (A419259), an SKI, improved cellular viability, the cell number and LDH release in 97Q cells in a dose-dependent manner (Fig. 3a–f, Supplementary Fig. 2). We also tested the effect of three other SKIs, SKI-1, PP2, and saracatinib. Treatment of the cellular models of SBMA with these compounds increased cell viability (Supplementary Fig. 3). To achieve selective inhibition of Src, we knocked down Src with siRNA. Silencing of Src increased cell viability and decreased LDH release in 97Q cells (Supplementary Fig. 4). We then investigated the effect of Src activation on 97Q cells (Fig. 3g–l). Transient overexpression of Src reduced cellular viability and the cell number and increased cellular damage, as reflected by the elevated release of LDH by both NSC34 and C2C12 cells stably expressing AR-97Q. Based on these results, Src plays a critical role in the pathogenesis of SBMA, and suppression of the Src pathway is potentially an effective therapy for SBMA.

**An SKI improves the phenotypes of SBMA mice**. We first investigated the optimal concentration of A419259 to suppress Src phosphorylation in the spinal cord and skeletal muscle of AR-97Q mice. To this end, we administered 0.25, 0.5 or 1 mg/kg/day of A419259 to AR-97Q mice once in three days for three times and anatomized them 2 h after the last administration to analyze the level of p-Src in tissues. The results showed that the effects of A419259 on Src phosphorylation was stronger at 0.5 or 1 mg/kg/ day compared with 0.25 mg/kg/day both in the spinal cord and skeletal muscle of AR-97Q mice. As the effect was similar between 0.5 and 1 mg/kg/day, we chose 0.5 mg/kg/day as the optimal dose. (Supplementary Fig. 5b, c). We measured the concentration of A419259 in the serum of AR-97Q mice 10 min after the injection of each dose of A419259. The concentrations of A419259 in the serum of the mice treated with 0.25 mg/kg/day A419259 were approximately half of those in the mice treated with 0.5 mg/kg/day A419259, whereas the serum concentrations

in the mice treated with 1.0 mg/kg/day A419259 were similar to those in the mice treated with 0.5 mg/kg/day of A419259, suggesting that the effect of Src kinase inhibition of A419259 is dose-dependent till 0.5 mg/kg/day (Supplementary Fig. 5d). These results support the findings in immunoblots: the expression levels of p-Src in the spinal cord and skeletal muscle of the mice treated with 0.5 mg/kg/day of A419259 are similar to those in the mice treated with 1.0 mg/kg/day (Supplementary Fig. 5b, c).

As for the frequency of administering A419259 to AR-97Q mice, we investigated the duration of Src inhibition with A419259 at 0.5 mg/kg/day by administrating the compound to mice once in three days for three times and analyzing the Src phosphorylation levels in tissues one, three or four days after the last injection. The p-Src was down-regulated both in the spinal cord and skeletal muscle until three days after the last injection, but tended to elevate four days after the last administration (Supplementary Fig. 5e–g).

To evaluate the possible off-targets of A419259, we investigated the phosphorylation levels of p38MAPK, JNK, IκBα, and p44/42 MAPK in the spinal cords and skeletal muscles of AR-97Q mice with and without A419259. We selected these molecules because p38MAPK and p44/42 MAPK are key regulators of cell survival and JNK and IκBα are involved in the pathogenesis of SBMA[17,18]. Injection of 0.5 mg/kg/day of A419259 once every three days for three times did not alter the phosphorylation levels of these molecules in the spinal cords and skeletal muscles of AR-97Q mice compared with those of vehicle-treated mice (Supplementary Fig. 6).

We intraperitoneally injected 6-week-old SBMA transgenic mice with 0.5 mg/kg/day A419259 once every three days to examine the effects of this SKI in vivo. The intraperitoneal administration of SKI to mice beginning at 6 weeks of age improved body weight, grip strength, and performance on the rotarod task and extended the lifespan of AR-97Q mice compared with vehicle-treated mice, although a reduced dose of A419259 did not ameliorate the neurological phenotype of AR-97Q mice (Fig. 4a–d and Supplementary Fig. 7). The administration of 0.5 mg/kg/day A419259 also attenuated muscle atrophy and improved the stride length of AR-97Q mice (Fig. 4e–g). On the other hand, administration of 0.5 mg/kg/day A419259 did not exert detectable effect on the motor phenotype of wild-type mice (Supplementary Fig. 8). We also investigated the effects of A419259 when administration was initiated at 8 weeks of age after the onset of neurological symptoms. The administration of A419259 to mice beginning at 8 weeks of age increased the body weight, grip strength, time on the rotarod and survival rate of AR-97Q mice. However, its effects on lifespan were weaker than on the group that received treatment beginning at 6 weeks of age. Specifically, the lifespans of the mice that received treatment beginning at 6 and 8 weeks were 39.4% and 24.2% longer, respectively, than those of vehicle-treated AR-97Q mice (Supplementary Fig. 9). Serum was collected from the mice at 13 weeks of age, and the serum levels of creatine kinase, aspartate

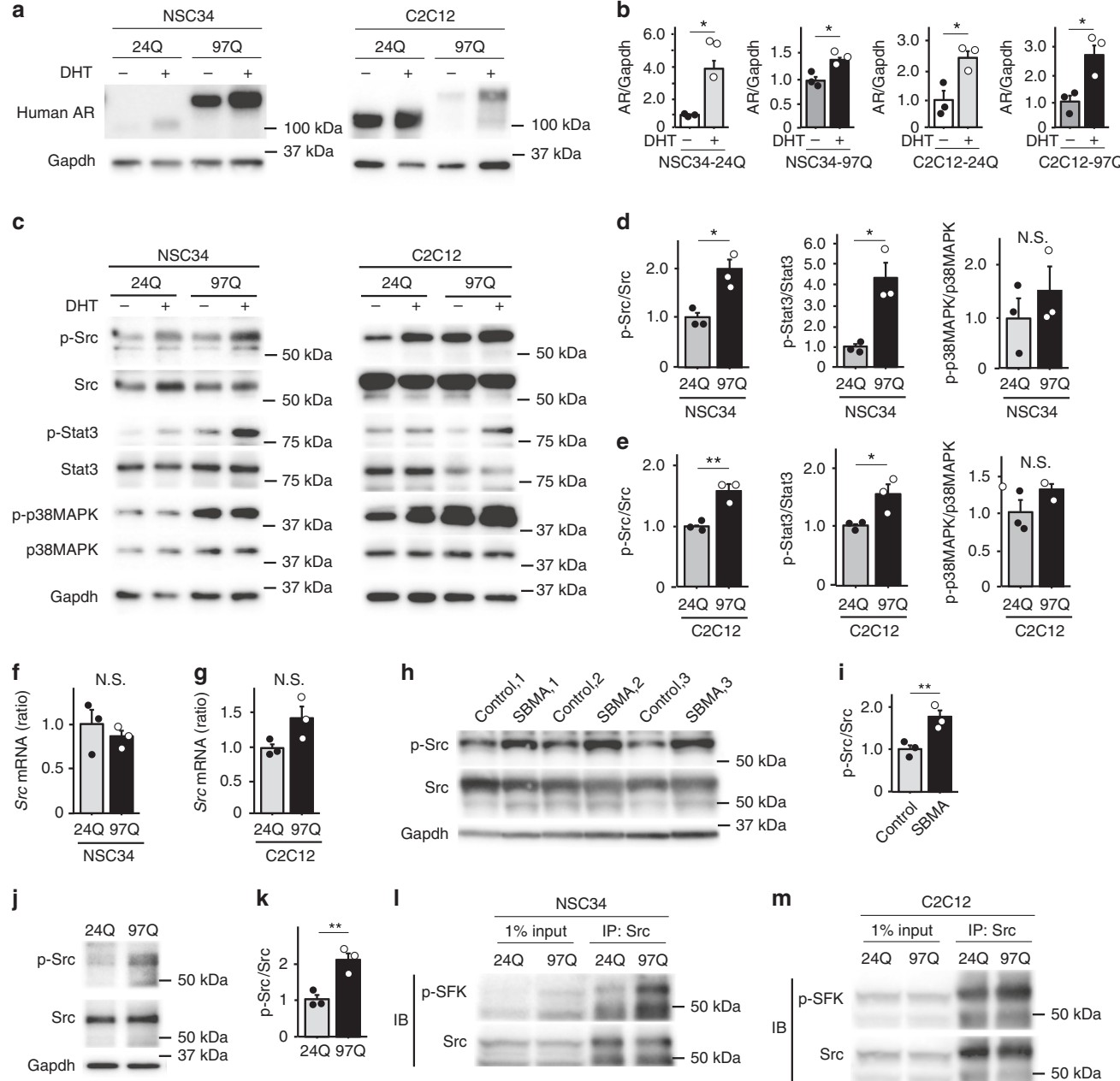

**Fig. 2** Levels of phosphorylated Src, Stat3 and p38MAPK in cellular models of SBMA. **a** Immunoblots showing the levels of the human AR protein in NSC34 and C2C12 cells stably expressing AR-24Q or AR-97Q that were treated with or without DHT. **b** Quantitative densitometry analysis of AR levels in NSC34 and C2C12 cells stably expressing AR-24Q or AR-97Q that were treated with or without DHT ($n = 3$ samples per group). **c–e** Immunoblots showing the levels of phosphorylated and total Src, Stat3, and p38MAPK proteins in AR-24Q cells and AR-97Q cells treated with or without DHT (**c**). Quantitative densitometry analysis of p-Src, p-Stat3 and p-p38MAPK levels in DHT-treated NSC34 (**d**) and C2C12 (**e**) cells stably expressing AR-24Q or AR-97 ($n = 3$ samples per group). **f, g** The mRNA levels of *Src* in DHT-treated NSC34 (**f**) or C2C12 (**g**) cells stably expressing AR-24Q or AR-97Q ($n = 3$ samples per group). **h, i** Immunoblots showing the levels of phosphorylated and total Src proteins in induced pluripotent stem cells (iPSCs) from patients with SBMA and healthy controls (**h**). Quantitative densitometry analysis of p-Src levels in iPSCs from patients with SBMA and healthy controls ($n = 3$ samples per group) (**i**). **j, k** Immunoblots showing the levels of phosphorylated and total Src proteins in Hu5/KD3 cells overexpressing AR-24Q or AR-97Q (**j**). Quantitative densitometry analysis of p-Src levels in Hu5/KD3 cells overexpressing AR-24Q or AR-97Q ($n = 3$ samples per group) (**k**). **l, m** NSC34 (**l**) and C2C12 (**m**) cell lysates were immunoprecipitated with a Src-specific antibody and immunoblotted with an antibody against Src family proteins. Error bars indicate the s.e.m. *$p < 0.05$ and **$p < 0.01$, unpaired two-sided *t*-test (**d–g**, **i**, **k**). DHT, dihydrotestosterone. N.S., not significant. Source data are provided as a Source Data file (**a–m**)

aminotransferase, alanine aminotransferase and blood urea nitrogen were measured to analyze the side effects of A419259. In addition, we assessed the levels of white blood cells, hemoglobin and platelets in whole blood samples obtained from 13-week-old mice. No abnormal values were observed in

A419259-treated AR-97Q mice, indicating that the intraperitoneal administration of SKI did not induce adverse systemic effects (Supplementary Fig. 10).

We next examined the effects of p-Src-targeted therapy on the expression of AR and Src in AR-97Q mice (Fig. 4h). Quantitative

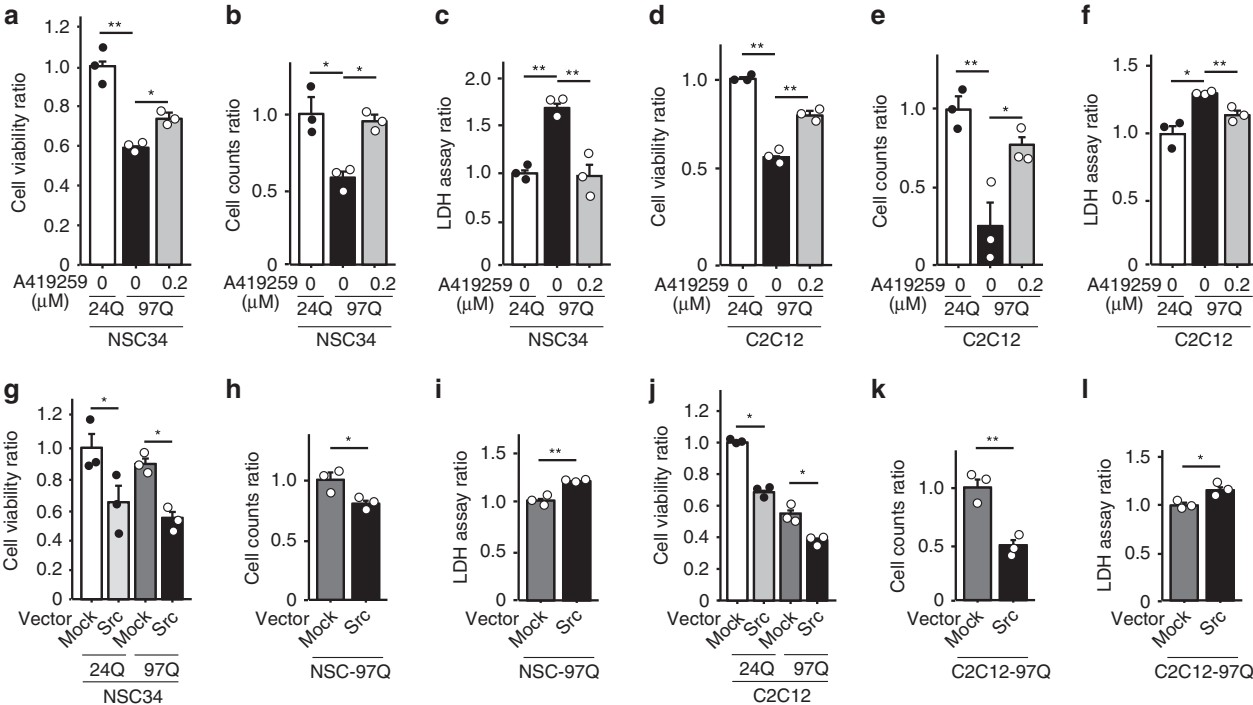

**Fig. 3** Src kinase inhibition improves the viability of cellular models of SBMA. **a–f** The viability, cell numbers and LDH release from NSC34 (**a–c**) and C2C12 cells (**d–f**) stably expressing AR-24Q or AR-97Q that were treated with or without A419259 are shown. **g** The viability of the NSC34 cells stably expressing AR-24Q or AR-97Q transfected with mock or Src plasmids. **h, i** The cell numbers (**h**) and LDH release (**i**) from the NSC34 cells stably expressing AR-97Q transfected with mock or Src plasmids are shown. **j** The viability of the C2C12 cells stably expressing AR-24Q or AR-97Q transfected with mock or Src plasmids. **k, l** The cell numbers (**k**) and LDH release (**l**) from C2C12 cells stably expressing AR-97Q transfected with mock or Src plasmids are shown. DHT was added to the cell culture medium in every assay. Quantitative analyses were performed on $n = 3$ samples per group. Error bars indicate the s.e.m. *$p < 0.05$ and **$p < 0.01$, unpaired two-sided $t$-test (**b, c, e, f, h, i, k, l**), ANOVA with Dunnett's test (**a, d**) or ANOVA with Tukey's test (**g, j**). Source data are provided as a Source Data file (**a–l**)

densitometry analysis revealed that A419259 did not alter the levels of either high-molecular-weight complexes or monomers of the AR protein in the spinal cords and skeletal muscles of AR-97Q mice (Fig. 4i, j). The phosphorylation levels of Src were suppressed in the spinal cords and skeletal muscles of A419259-treated AR-97Q mice compared with those in vehicle-treated AR-97Q mice. We performed immunohistochemistry on the spinal cords and skeletal muscles of wild-type, vehicle-treated AR-97Q and A419259-treated AR-97Q mice to investigate the histopathological changes underlying the beneficial effects of the A419259 treatment on AR-97Q mice. In samples subjected to immunohistochemical staining with the 1C2 antibody, which specifically recognizes expanded polyglutamine residues, the number of 1C2-positive cells in the spinal cords and skeletal muscles of vehicle-treated and A419259-treated AR-97Q mice was not significantly different (Fig. 5a–d). Immunohistochemical staining of sections from the anterior horn of the spinal cord with an antibody against choline acetyltransferase (ChAT) showed motor neuron atrophy in vehicle-treated AR-97Q mice compared with wild-type mice, and the motor neuron size increased in response to the A419259 treatment (Fig. 5e, f). Immunohistochemistry using an antibody against neurofilament heavy chain also revealed that the size of motor neurons was decreased in vehicle-treated AR-97Q mice compared with wild-type mice, and that the size were restored by the treatment with A419259 (Supplementary Fig. 11). Hematoxylin and eosin (HE) staining revealed skeletal muscle fiber atrophy in vehicle-treated AR-97Q mice compared with wild-type mice, and the A419259 treatment improved amyotrophy in AR-97Q mice (Fig. 5g, h). We also examined the effects of other Src inhibitors in vivo, but intraperitoneal injection of SKI-1 or PP2

every other day for three times did not suppress Src phosphorylation in the spinal cord and skeletal muscle of AR-97Q mice (Supplementary Fig. 12). Limited bioavailability may underlie the ineffectiveness of SKI-1 and PP2 in vivo, given that these compounds mitigate the mutant AR toxicity in cellular models (Supplementary Fig. 3).

**p130Cas is the effector of the Src pathway in SBMA**. We examined the phosphorylated and total levels of molecules downstream of Src[19,20], including p130Cas, Stat3, p38MAPK, and Akt, by immunoblotting the spinal cord and skeletal muscle from AR-97Q mice treated with or without A419259 for 4 weeks to identify effectors of the Src pathway (Fig. 6a). According to the results of the densitometry analysis, the ratios of phosphorylated to total p130Cas and Stat3 proteins were significantly decreased in the spinal cords and skeletal muscles of A419259-treated AR-97Q mice compared with vehicle-treated AR-97Q mice, suggesting that, among the effectors of Src in AR-97Q mice, A419259 primarily targets p130Cas and Stat3 (Fig. 6b, c). We then examined the impact of A419259 on the cellular models of SBMA by measuring the levels of phosphorylated and total proteins that act downstream of Src. Immunoblotting and densitometry analyses revealed significantly diminished p-Src and p-p130Cas levels in 97Q cells following treatment with A419259 (Fig. 6d–f). Furthermore, Src overexpression using pcDNA3 c-SRC (91–536) plasmid markedly activated p130Cas in both the NSC34 and C2C12 cell lines (Fig. 6g–i). Based on these results, we concluded that Src kinase directly influences the phosphorylation of p130Cas during the pathogenesis of SBMA.

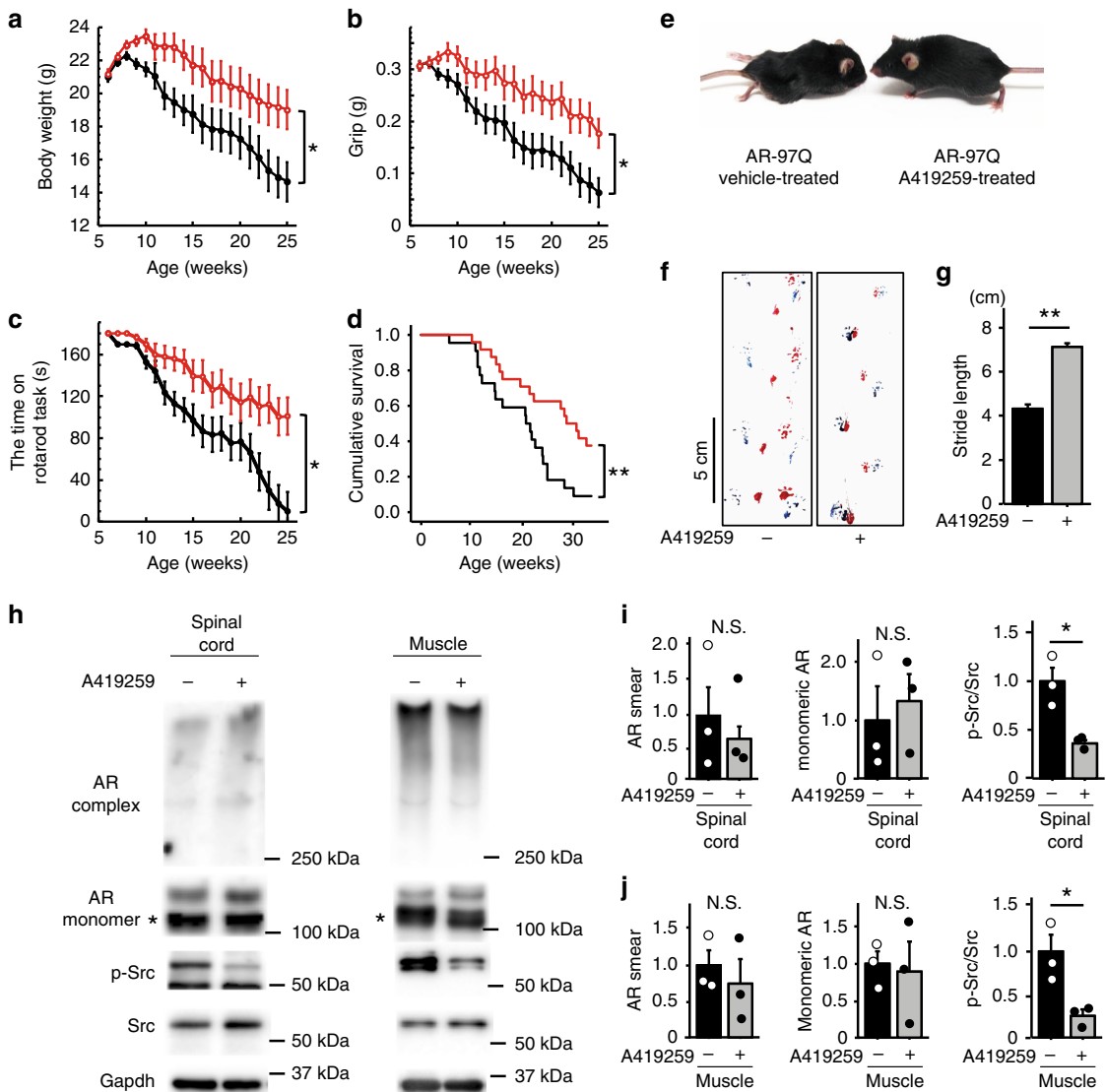

**Fig. 4** A419259 improves the neuromuscular phenotype of the mouse model of SBMA. **a–d** Body weight (**a**), grip strength (**b**), rotarod performance (**c**) and survival rate (**d**) of A419259-treated ($n = 24$) and vehicle-treated AR-97Q mice ($n = 22$). All parameters improved after treatment with A419259 (*$p < 0.05$ at 13 weeks, as determined by the unpaired two-sided $t$-test for body weight, grip power and rotarod performance; **$p < 0.01$ by the log-rank test). **e** Muscle atrophy and kyphosis in 13-week-old A419259- and vehicle-treated AR-97Q mice. **f** Footprints of 13-week-old AR-97Q mice. The front paws are indicated in red, and the hind paws are indicated in blue. **g** The length of steps taken by 13-week-old AR-97Q mice ($n = 3$ animals per group). The columns show the average step length of the hind paw. **h–j** Immunoblots for AR, p-Src, and Src in the spinal cord and skeletal muscle of 13-week-old AR-97Q mice treated with or without A419259 (**h**). Quantitative analyses were performed using densitometry ($n = 3$ animals per group) (**i**, **j**). Black lines indicate vehicle-treated mice and red lines indicate A419259-treated mice (**a–d**). Error bars indicate the s.e.m. *$p < 0.05$ and **$p < 0.01$, unpaired two-sided $t$-test (**g**, **i**, **j**). N.S., not significant. *, Non-specific bands. Source data are provided as a Source Data file (**a–d**, **g–j**)

We then analyzed the levels of phosphorylated and total p130Cas in NSC34 and C2C12 cells stably expressing AR-24Q and AR-97Q (Fig. 7a). The phosphorylation of p130Cas was significantly increased in DHT-treated NSC34 and C2C12 cells stably expressing AR-97Q compared with DHT-treated cells expressing AR-24Q, according to the densitometry analysis (Fig. 7b). Moreover, p130Cas was activated in the spinal cords of AR-97Q mice at 6, 9, and 13 weeks of age and in the skeletal muscles of AR-97Q mice at 6 and 9 weeks of age compared with control mice (Fig. 7c–e). We also observed Src activation in the spinal cords and skeletal muscles of AR-97Q mice at these ages using Bio-Plex arrays (Fig. 1a). Immunohistochemical staining using an antibody against p-p130Cas showed increased immunoreactivity in the cytoplasm of motor neurons in the spinal cords and skeletal muscles of vehicle-treated AR-97Q mice at

9 weeks of age compared with wild-type mice; these effects were attenuated by the A419259 treatment (Fig. 7f, g).

**Phosphorylation of p130Cas affects the pathogenesis of SBMA.** The p130Cas adaptor protein, also known as BCAR1, belongs to the CAS family of scaffold proteins that participate in cell migration, cell cycle and apoptosis[21,22]. To confirm that the upregulated phosphorylation of p130Cas affects the pathogenesis of SBMA, we first down-regulated p130Cas using siRNA. Silencing of p130Cas increased cell viability and the cell number and decreased LDH release in 97Q cells (Fig. 8a–c, Supplementary Fig. 13). Overexpression of p130Cas decreased cell viability and the cell number and increased LDH release in 97Q cells (Fig. 8d–f), suggesting that the suppression of p-p130Cas reduced

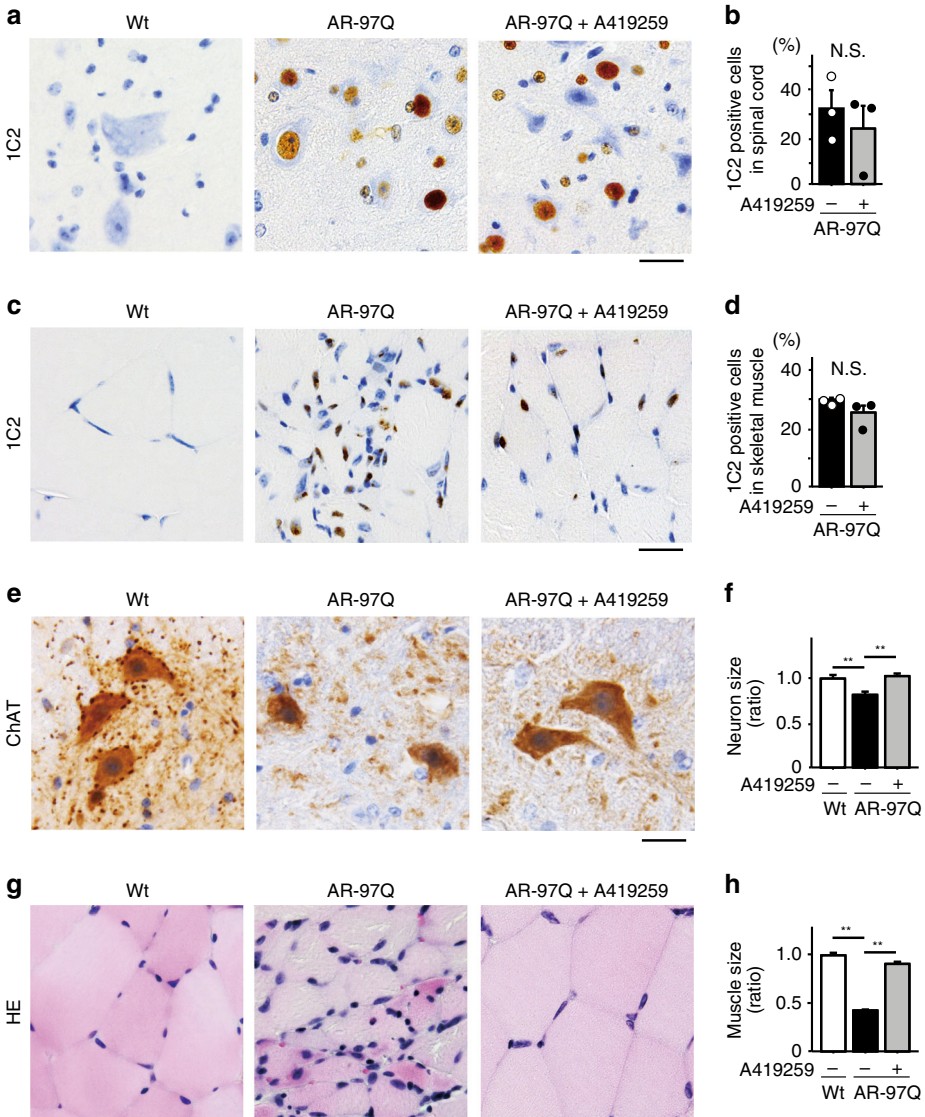

**Fig. 5** Effect of A419259 on the histopathology of AR-97Q mice. **a–d** Immunohistochemical staining of the spinal cord (**a**, **b**) and skeletal muscle (**c**, **d**) from 13-week-old mice with an antibody against polyglutamine (1C2) and quantitative analysis of the staining. **e**, **f** Immunostaining for ChAT in spinal cord samples from 13-week-old mice. **g**, **h** HE staining of skeletal muscle (**g**) and quantitation of the muscle fiber size (**h**) in 13-week-old mice. Quantitative analyses were performed on sections from $n = 3$ animals per group. Error bars indicate s.e.m. Statistical analyses were performed using the unpaired two-sided $t$-test (**b**, **d**). **$p < 0.01$, ANOVA with Dunnett's test (**f**, **h**). N.S., not significant. Scale bars: 25 μm (**a**, **c**, **e**, **g**). Source data are provided as a Source Data file (**b**, **d**, **f**, **h**)

the neuromuscular degeneration associated with SBMA and that activation of this protein exerts a causative effect on disease pathogenesis. Moreover, bosutinib, a p130Cas inhibitor[23], increased cell viability and attenuated LDH release of 97Q cells (Supplementary Fig. 14). Suppression of p130Cas down-regulated cleaved caspase 3 and overexpression of p130Cas increased cleaved caspase 3 levels in 97Q cells, suggesting that p130Cas induces cellular damage by activating apoptotic pathway (Supplementary Fig. 15). We overexpressed p130Cas in NSC34 cells and investigated the levels of phosphorylated and total p38MAPK and Stat3 to verify the relationships among p130Cas, p38MAPK and Stat3 (Supplementary Fig. 16a). According to the results of the densitometry analysis, p130Cas overexpression did not alter the levels of either p-p38MAPK or p-Stat3 in NSC34 cells (Supplementary Fig. 16b). Similar findings were obtained from C2C12 cells transfected with the p130Cas vector (Supplementary Fig. 16c, d).

**Hyperactivation of AR induces Src phosphorylation**. To explore the relationship between Src and AR, we overexpressed a mock vector or the Src vector in NSC34 and C2C12 cells stably expressing AR-24Q or AR-97Q (Supplementary Fig. 17a, b). Src overexpression did not alter the level of the AR protein in either cell line (Supplementary Fig. 17c). We further examined the levels of p-Src and Src in NSC34 cells stably expressing AR-24Q (Supplementary Fig. 18a). Src phosphorylation was elevated even in the NSC34 cells that stably expressed AR-24Q (Supplementary Fig. 18b); a similar finding was obtained from C2C12 cells (Supplementary Fig. 18c, d). Based on these findings, upregula-tion of native AR function activates the Src pathway, whereas Src has virtually no effect on the metabolism of the pathogenic AR protein. We further investigated the mechanisms of Src activation in the pathogenesis of SBMA. C-terminal Src kinase (Csk) is the major regulator of Src kinase, which catalyzes the phosphoryla-tion of Src at Tyr527 that suppresses the auto-phosphorylation of

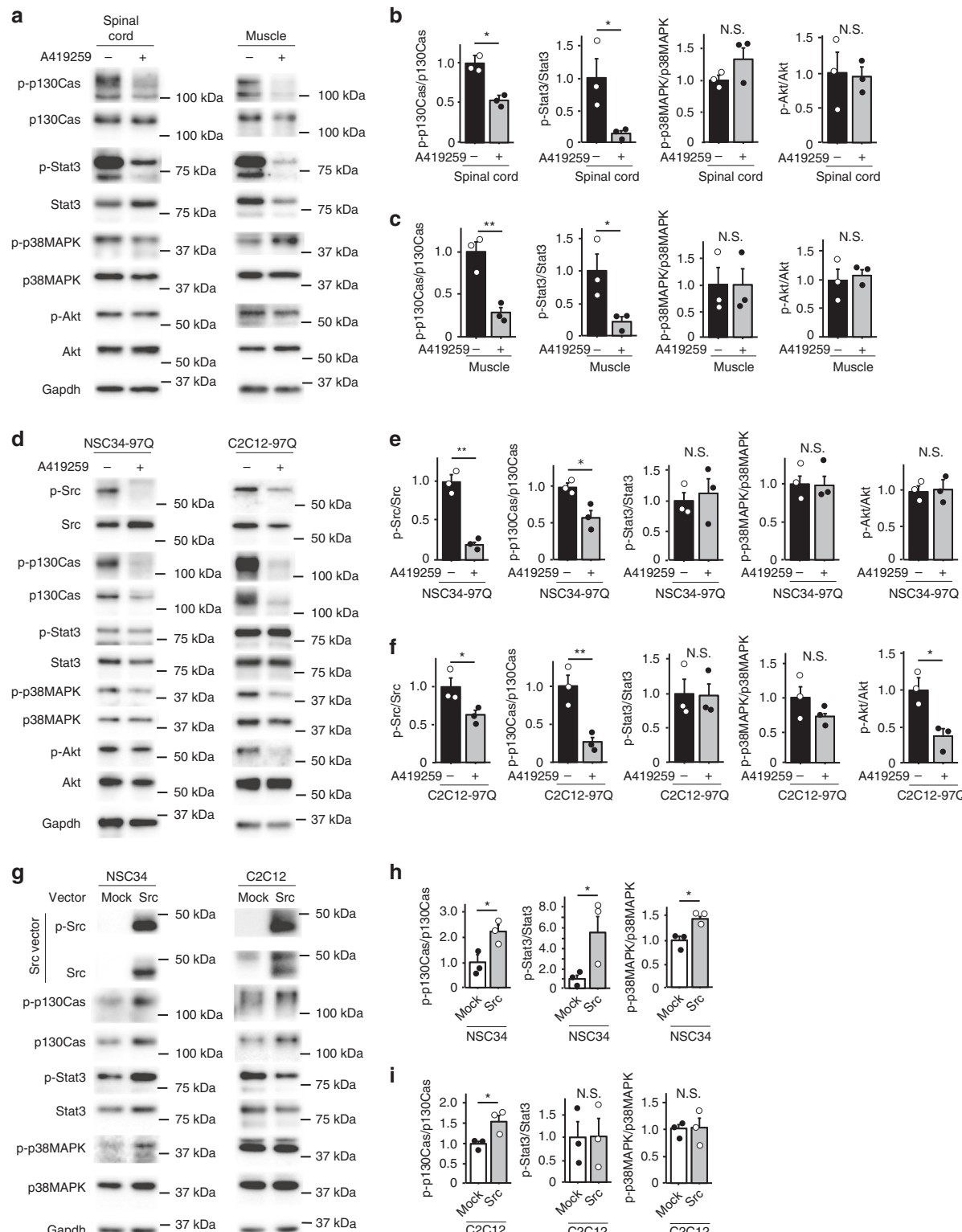

**Fig. 6** Downstream targets of Src in mouse and cellular models of SBMA. **a–c** Immunoblots showing the levels of phosphorylated and total p130Cas, Stat3, p38MAPK, and Akt proteins in the spinal cord and skeletal muscle of 9-week-old AR-97Q mice treated with or without A419259 (**a**). Quantitative analyses were performed using densitometry (**b**, **c**). **d–f** Immunoblots showing the levels of the phosphorylated and total Src, p130Cas, Stat3, p38MAPK, and Akt proteins in DHT-treated NSC34 and C2C12 cells stably expressing AR-97Q with or without 0.2 µM of A419259 administration (**d**). Quantitative analyses were performed using densitometry (**e**, **f**). **g–i** Immunoblots showing the levels of the phosphorylated and total Src, p130Cas, Stat3, and p38MAPK proteins in the NSC34 and C2C12 cells transfected with mock or Src plasmids (**g**). Quantitative analyses were performed using densitometry (**h**, **i**). Quantitative analyses were performed on samples from $n = 3$ animals per group. Error bars indicate the s.e.m. Statistical analyses were performed using the unpaired two-sided $t$-test (**b**, **c**, **e**, **f**, **h**, **i**). *$p < 0.05$ and **$p < 0.01$. N.S., not significant. Source data are provided as a Source Data file (**a–i**)

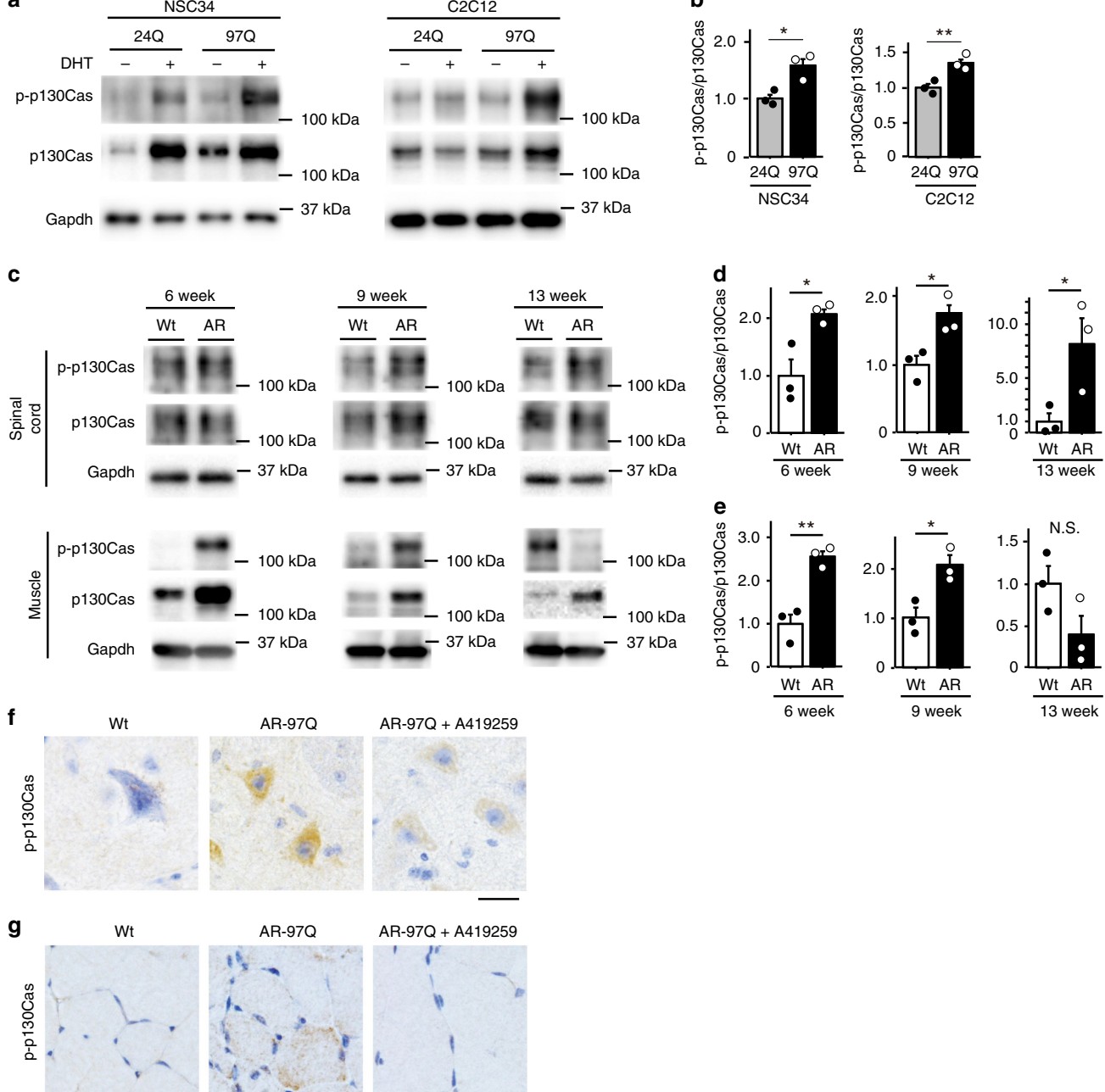

**Fig. 7** p130Cas is activated in cellular and mouse models of SBMA. **a**, **b** Immunoblots showing the levels of the phosphorylated and total p130Cas proteins in NSC34 cells and C2C12 cells stably expressing AR-24Q or AR-97Q that were treated with or without DHT (**a**). Quantitative analyses were performed using densitometry ($n = 3$) (**b**). **c–e** Immunoblots showing levels of the p-p130Cas and p130Cas proteins in the spinal cord and skeletal muscle of wild-type and AR-97Q mice at 6, 9 and 13 weeks of age (**c**). Quantitative analyses were performed using densitometry ($n = 3$) (**d**, **e**). **f**, **g** Immunohistochemistry for p-p130Cas in spinal cord (**f**) and skeletal muscle (**g**) samples from 6-week-old mice. Error bars indicate the s.e.m. Statistical analyses were performed using the unpaired two-sided $t$-test (**b**, **d**, **e**). *$p < 0.05$ and **$p < 0.01$. Wt, wild-type; AR, AR-97Q; N.S., not significant. Scale bars: 25 μm (**f**, **g**). Source data are provided as a Source Data file (**a–e**)

the kinase at Tyr416 and thereby inactivates Src[24,25]. In contrast, the dephosphorylation of Src at Tyr527 catalyzes the intermolecular phosphorylation of Tyr416[26]. We thus examined the expression level of Csk in 24Q or 97Q cells using western blotting and RT-PCR. However, the mRNA and protein level of Csk in 97Q cells were elevated compared with those in 24Q cells (Supplementary Fig. 19a-c), indicating that the upregulated Src phosphorylation at Tyr416 in SBMA was not mediated by Csk.

SKI competitively inhibited ATP activity and reduced the autophosphorylation of Src at Tyr416 in the pathophysiology of SBMA, as shown in Fig.6. We thus examined the impact of the interaction of Src with AR, because several reports showed that the direct association of AR with Src triggers the autophosphorylation of Src[27,28]. We confirmed that a Src activator MLR1023 activates the phosphorylation of exogenous Src (pcDNA3 c-SRC (91–536) plasmid) in NSC34 cells

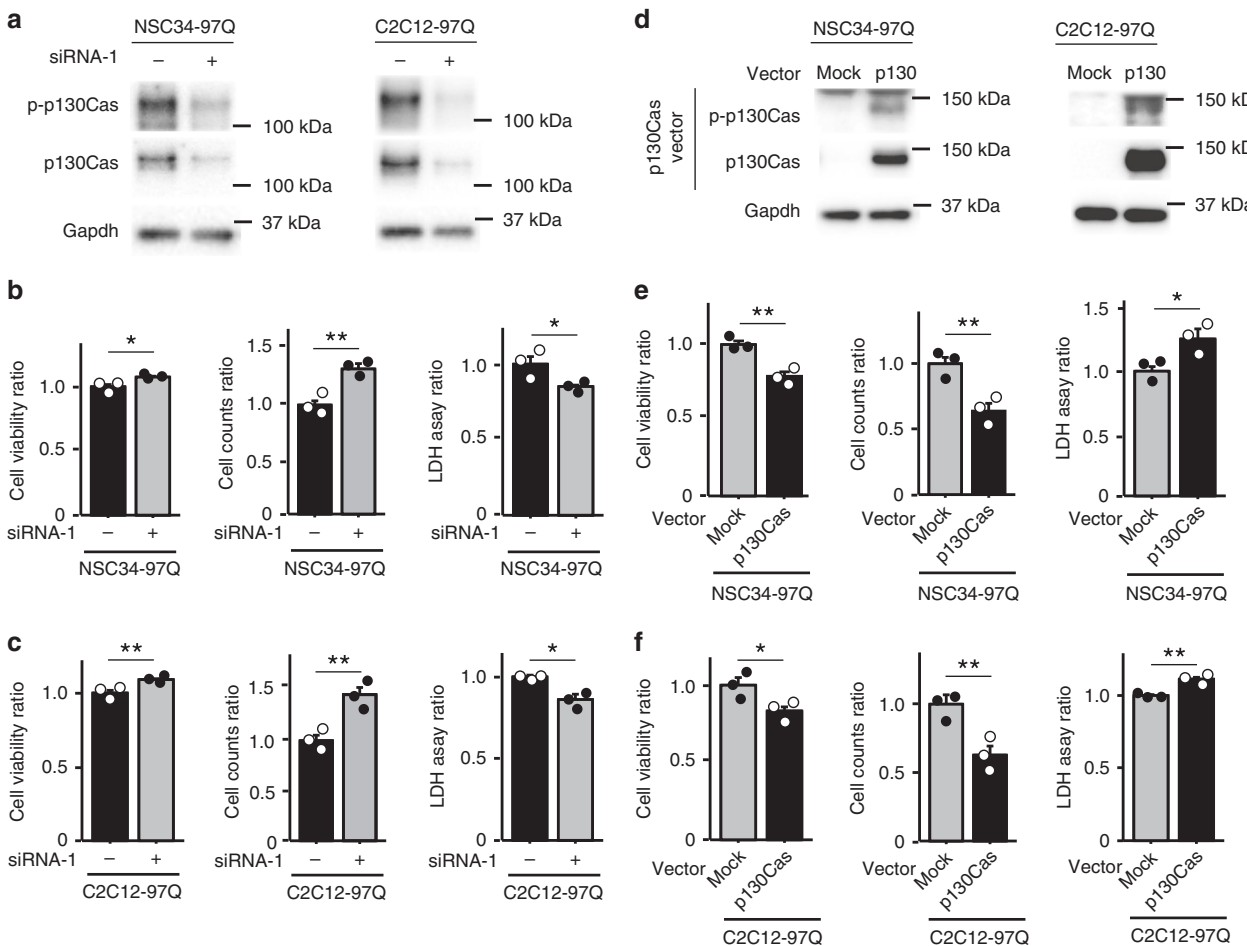

**Fig. 8** Phosphorylation of p130Cas exerts toxic effects on cellular models of SBMA. **a** Immunoblots showing the levels of the p-p130Cas and p130Cas proteins in DHT-treated NSC34 and C2C12 cells stably expressing AR-97Q and transfected with the mock or p130Cas siRNA-1. **b, c** The viability, cell numbers and LDH release from NSC34 (**b**) and C2C12 cells (**c**) are shown ($n = 3$). **d** Immunoblots showing the levels of phosphorylated and total p130Cas proteins in DHT-treated NSC34 and C2C12 cells stably expressing AR-97Q and transfected with the mock or p130Cas vector. **e, f** Viability, cell numbers and LDH release from NSC34 (**e**) and C2C12 (**f**) cells are shown ($n = 3$). Error bars indicate the s.e.m. Statistical analyses were performed using the unpaired two-sided $t$-test (**b, c, e, f**). *$p < 0.05$ and **$p < 0.01$. Source data are provided as a Source Data file (**a–f**)

(Supplementary Fig. 19d). The SH3 domain of Src binds to 372–385 amino acid residues in human AR[29], which correspond to the DNA sequence of 1345–1386 in the AR-97Q vector. We conducted site-specific mutagenesis and created the AR construct with a deletion of 1345–1386 (deleted AR-97Q vector), and co-transfected NSC34 cells with the Src vector and the full-length AR-97Q or deleted AR-97Q vector and treated the cells with DHT. To assess the interaction of Src with deleted AR-97Q, we immunoprecipitated lysates using an anti-Src antibody and immunoblotted the immunocomplexes with an AR antibody. We detected the strong association of Src with full-length AR-97Q, whereas a much weaker association was observed in cells transfected with the Src vector and deleted AR-97Q vector (Supplementary Fig. 19e). We then examined the phosphorylation level of Src in the DHT-treated cells using immunoblotting, which revealed decreased Src phosphorylation in cells transfected with the Src vector and deleted AR-97Q vector, compared with the cells transfected with the Src vector and full length AR-97Q vector (Supplementary Fig. 19f, g). We next performed experiments using the SH3 domain-deleted Src (SH3·Src vector: pcDNA3 c-SRC (151–536) plasmid). The results showed that the Src activator MLR1023 is capable of phosphorylating SH3·Src in NSC34 cells (Supplementary Fig. 19h). To assess the interaction of SH3·Src with AR-97Q, we transfected NSC34 cells

stably expressing AR-97Q with the full-length Src vector or SH3·Src vector, treated the cells with DHT, immunoprecipitated the cell extract using the Src antibody and immunoblotted the proteins with the AR antibody. We detected the strong association of AR-97Q with the full-length Src, while there was little association between SH3·Src and AR-97Q (Supplementary Fig. 19i). We then evaluated the phosphorylation levels of the full-length and SH3·Src in DHT-treated 97Q cells. The results demonstrated that Src phosphorylation was attenuated by the SH3 domain deletion (Supplementary Fig. 19j, k). Furthermore, to assess the effect of androgen on the interaction of Src with the pathogenic AR, we transfected NSC34 cells stably expressing AR-97Q with the full-length Src vector and treated the cells with ethanol or DHT. We then immunoprecipitated the cell extract using the Src antibody and immunoblotted the proteins with the antibody against AR. The treatment of 97Q cells with DHT upregulated the expression levels of AR, being consistent with the results in Fig. 2a. We detected the strong association of Src with AR-97Q in the cells treated with DHT, whereas Src and AR-97Q were hardly associated with each other without DHT treatment (Supplementary Fig. 19i). Taken together, these results suggest that the interaction between Src and AR, which is strengthened by DHT, plays an essential role in the activation of Src pathway in the pathogenesis of SBMA.

## Discussion

In the present study, we identified the Src signaling pathway as a mediator of the neuromuscular toxicity of the polyglutamine-expanded AR protein and discovered that SKI is a potential therapy for SBMA. Src kinase is activated in a large number of human cancers, including breast, colon, brain, and prostate cancer, and regulates the progression[30]. In addition, Src kinase is expressed at high levels in the brain and in neural tissues[31]. The present study thus provided further insight into the commonalities between neurodegeneration and cancer, which have been intensively discussed with regard to various disorders[32,33].

The relationship between AR and Src has been well studied in androgen-independent prostate cancer, in which AR becomes hypersensitive to androgen[34,35]. According to recent clinical findings, AR activity is regulated not only by the presence of low levels of androgens but also by cross-talk with the Src kinase cascade, implicating activated Src as an important mediator of AR signaling[36,37]. Ligand-induced activation of AR upregulates the autophosphorylation of Src, whereas culture of a prostate tumor cell line under androgen ablation conditions dramatically attenuates Src autophosphorylation[38]. In addition, androgen induces the assembly of a ternary complex comprising AR and Src, thereby triggering stronger stimulation of Src activity[38]. Furthermore, inhibition of AR expression using an siRNA decreased Src activation but not total Src expression in a cellular model of prostate cancer, indicating that AR regulates Src activation but not its expression[39]. In contrast, Src silencing with siRNA does not alter AR expression[39]. These findings are consistent with our results showing that the polyglutamine-expanded AR activated Src but not mRNA or protein level of Src, although Src had no effect on the steady-state levels or aggregation of the pathogenic AR. Based on these evidence, the presence of elongated polyglutamine tracts results in the excessive intensification of the function of the native protein during the pathogenesis of polyglutamine diseases, including SBMA[40,41]. In the present study, Src phosphorylation was increased in NSC34 and C2C12 cells stably expressing AR-24Q, albeit to a lesser extent than AR-97Q, suggesting that the hyperactivation of native AR function resulted in elevated Src activity and that the expression of polyglutamine proteins increased Src activation. Moreover, the interaction of AR and Src was crucial to activate Src pathway, and this is a prominent mechanism underlying Src activation in the pathogenesis of SBMA.

The Src family kinases include nine non-receptor protein tyrosine kinases that possess conserved and similar structures and functions[42,43]. These kinases have been implicated in the progression and metastasis of various cancers and have essential roles in cell adhesion, proliferation, survival, and invasion[30]. While the elevation of Src expression is often modest, several fold increases in Src activity have been observed in several cancers, suggesting the importance of Src phosphorylation, rather than an increase in Src protein levels. The Src pathway has also been implicated in several disorders of the nervous system. For instance, Src kinase regulates vascular permeability, and the suppression of Src activity with an SKI improves cerebral perfusion[44]. Moreover, Src has been implicated in glutamate toxicity, a major contributor to the neurodegenerative process; Src knockdown rescues neural cells from glutamate-induced cell death[45]. Fyn, another Src family kinase, is involved in synaptic plasticity and is activated by Aβ oligomers, leading to synaptic and cognitive impairments in AD transgenic mice[46,47]. Targeting Fyn with an SKI reverses memory deficits in AD mouse models[48]. A recent report showed beneficial effects of Src/c-Abl inhibitors on iPSC-derived motor neurons from patients with ALS and on mice expressing mutant superoxide dismutase 1[49]. SKIs have been shown to be well tolerated in humans, and the SKIs dasatinib and bosutinib have been used to

treat chronic myelogenous leukemia and acute lymphocytic leukemia. Saracatinib, another SKI, was tolerated by patients in phase 2 clinical trials of cancer treatments[50,51]. Because the Src pathway has been implicated in various neurodegenerative diseases, SKIs appear to be promising candidate therapeutics for the treatment of neurodegeneration.

Src has been shown to be involved in many different pathways, including cellular responses to tyrosine kinase receptors, G protein-coupled receptors, channel activity and stress, all of which relate to the complexity of Src functions. The most critical activity of the Src pathway is the phosphorylation of downstream targets such as p130Cas, Stat3, p38MAPK and Akt[19]. In the present study, p130Cas phosphorylation was substantially decreased when an SKI was administered to cellular and mouse models of SBMA. Moreover, decreased levels of phosphorylated p130Cas improved the viability of cellular models of SBMA, whereas the augmentation of p-p130Cas by overexpressing its vector reduced the viability of cellular models of SBMA, suggesting that p130Cas activation is directly related to the pathogenesis of SBMA. The p130Cas protein is involved in the development and progression of several human cancers, including hematological malignancies, prostate cancer and colorectal cancer[52]. The p130Cas protein binds to Smad3 and prevents its phosphorylation by the transforming growth factor-β (TGF-β) receptor, resulting in cell cycle arrest[53]. Moreover, overexpression of p130Cas in mammary epithelial cells not only decreases the activation of Smad2/3 by the TGF-β receptor but also increases the coupling of TGF-β to p38MAPK signaling, resulting in the activation of this kinase[54]. The phosphorylation of p130Cas also increases JNK activation via the p130Cas-CRK signaling complex[55,56]. These observations are consistent with previous reports showing the suppression of the nuclear translocation of phosphorylated Smad2/3 and activation of the JNK pathway in the spinal motor neurons of the mouse model of SBMA[17,57]. Furthermore, in the present study, the level of phosphorylated p130Cas was increased in the mouse model of SBMA before disease onset and in the cellular models of SBMA. These alterations in the p38MAPK pathway may be mediated by activated p130Cas.

## Methods

**Bio-Plex phosphoprotein assay.** Protein lysates were prepared using cell lysis buffer (Bio-Rad Laboratories, Hercules, CA, USA). We measured the levels of the following molecules: p-Src (Tyr416), p-Stat3 (Tyr705), p-p38MAPK (Thr180/Tyr182), p-c-Jun (Ser63), p-IGF-1R (Tyr1131), p-IRS-1 (Ser636/Ser639), p-Akt (Ser473), p-IkBα (Ser32/Ser36), p-p70s6k (Thr389), p-Erk1/2 (Thr202/Tyr204, Thr185/Tyr187), p-Smad2 (Ser465/Ser467), p-GSK (Ser21/Ser9), p-c-Abl (Tyr245), p-JNK (Thr183/Tyr185), p-VEGFR2 (Tyr1175), p-mTOR (Ser2448), and p-HSP27 (Ser78). Assays designed to quantify the levels of these 17 phosphorylated proteins were performed using a Bio-Plex Pro™ Cell Signaling Assay kit (Bio-Rad Laboratories), according to the manufacturer's instructions. Briefly, 50 μL of lysate (600 μg of protein/mL) from the mouse spinal cord or skeletal muscle were placed in each well of a 96-well plate coated with magnetic beads conjugated with capture antibodies, and the plate was incubated overnight at room temperature on a platform shaker at $2 \times g$. After an incubation with detection antibodies for 30 min at room temperature, data were acquired and analyzed using the Luminex®200 xPONET® 3.1 system (Merck Millipore).

**Cell culture and transfection.** Mouse NSC34 motor neuron-like cells (kindly provided by N.R. Cashman, University of British Columbia, Vancouver, Canada), mouse C2C12 myoblast cells (ECACC number: 91031101) (#EC91031101-G0, DS Pharma Biomedical, Osaka, Japan) and human myogenic cell clone Hu5/KD3 (a kind gift from Naohiro Hashimoto, National Center for Geriatrics and Gerontology, Aichi, Japan)[58] were cultured in a humidified atmosphere of 95% air/5% $CO_2$ in a 37 °C incubator. NSC34 and C2C12 cells were maintained in Dulbecco's Modified Eagle's Medium (DMEM) supplemented with 10% fetal bovine serum (FBS). Hu5/KD3 cells were cultured in DMEM supplemented with 20% FBS and 2% Ultroser G (Biosepra, Cergy-Saint-Christophe, France). The plasmids were transfected using OPTI-MEM (Gibco, Germany) and Lipofectamine 2000 (Invitrogen, Carlsbad, CA, USA), according to the manufacturer's instructions. Stable clones expressing AR-24Q or AR-97Q were established by selection with the

antibiotic G418 (Sigma-Aldrich, MO, USA) (0.4 mg/ml final concentration). NSC34 cells were differentiated in DMEM for 48 h. Fetal horse serum (2%) was added to the medium for 48 h to differentiate the C2C12 cells. Hu5/KD3 cells were differentiated in DMEM consisting of Insulin-Transferrin-Selenium (Invitrogen) and 2% of FBS for 24 h[59].

**Human iPSC-derived neurons.** Human iPSCs were established by the transduction of *OCT4, SOX2, KLF4, L-MYC, LIN28,* and *shTP53* by episomal vectors into the human dermal fibroblasts[60,61] of three SBMA patients (46-, 33-, and 38-year-old males) and three healthy controls (20-, 21-, and 24-year-old males). iPSCs were maintained on mitomycin-C-treated SNL murine fibroblast feeder cells in standard hESC medium, and were used for motor neuron differentiation[61]. iPSC-derived motor neuron progenitors were plated on poly-L-ornithine (Sigma-Aldrich) and Laminin (Thermo Fisher Scientific, Waltham, MA, USA)-coated dishes, cultured in motor neuron medium (MNM) consisting of media hormone mix (MHM)(KBM Neural Stem, Kohjin Bio, Japan)[61,62] with supplements for 4 weeks in the presence of DHT (Sigma-Aldrich), and used for the analysis.

**Plasmid constructs and siRNA.** The pCR3.1–AR–24Q and pCR3.1–AR–97Q plasmids were cloned using QIAfilter Plasmid Midi kit (Qiagen, Hilden, Germany)[63]. The pcDNA3 c-SRC (91–536) plasmid (Addgene plasmid # 42206)[64] and pcDNA3 c-SRC (151–536) plasmid (Addgene plasmid # 42207)[64] were gifts from Robert Lefkowitz, and the pEBG p130Cas plasmid (Addgene plasmid # 15001)[65] was a gift from Raymond Birge. The following oligonucleotide siRNA duplexes were synthesized by Invitrogen and transfected into NSC34 and C2C12 cells to knockdown p130Cas (p130Cas siRNA-1): sense sequence, GGGUCAAGGAGCU AGGCCATT; antisense sequence, UGGCCUAGCUCCUUGACCCTT. Another siRNA to knockdown p130Cas (p130Cas siRNA-2) were p130Cas siRNA (sc-36 142) (Santa Cruz Biotechnology, Texas, USA). To knockdown Src, we used c-Src siRNAs (sc-29859) (Santa Cruz Biotechnology). We used the Stealth RNAi negative control duplex (Invitrogen) as the control siRNA. NSC34 and C2C12 cells were transfected with the siRNA oligonucleotide duplex using Lipofectamine 2000 (Invitrogen), according to the manufacturer's instructions.

**Cell viability, cell counts, and toxicity assays.** Cell viability assays were performed using WST-8 (Roche Diagnostics, Mannheim, Germany) according to the manufacturer's instructions[18]. Cells were cultured in 24-well plates. Twenty-four hours after treatment with the indicated concentrations of A419259 (Tocris Bioscience), SKI-1 (Abcam, Cambridge, UK), PP2 (Cayman Chemical, MI, USA), saracatinib (Santa Cruz) or bosutinib (abcam), cells were incubated with the WST-8 substrate for 2–3 h, after which the absorbance of the wells was measured at 450 nm using a plate reader (2300 EnSpire™, PerkinElmer, Yokohama, Japan). The number of live cells was determined using a Countess cell counter (Invitrogen) after cultures were stained with trypan blue. Toxicity assays were performed using the Cytotoxicity Detection Kit PLUS (Roche Diagnostics, Indianapolis, IN). Twenty-four hours after treating the cells, 100 μL of the medium were removed from the plate and used in the assay. The medium was incubated with the substrate for 15 min and spectrophotometrically assayed at 490 nm using a plate reader. To activate the Src vector, we used MLR1023 (#4582) (Tocris Bioscience, Bristol, UK).

**Animals.** AR-97Q (Line #7–8) mice were bred and maintained in our institute of laboratory animals[66,67]. The animals were backcrossed for at least 15 generations to C57BL/6 mice before use in the present study. The mice were genotyped by PCR amplification using DNA extracted from the tail[66] with the primers listed in Supplementary Table 1. In the experiments, A419259 trihydrochloride was intraperitoneally administered at concentrations of 0.5 mg/kg/day beginning at 6 or 8 weeks of age until the end of the analysis, unless indicated otherwise. The litters were randomly allocated to A419259-treated or vehicle-treated groups. A419259 was dissolved in distilled $H_2O$, which was used as the vehicle. Only male mice were used in this study. For pathological and biochemical analyses, A419259 was administered to the mice beginning at 6 weeks of age. SKI-1 and PP2 were dissolved in dimethyl sulfoxide.

**Hematological analyses of mice.** We collected blood from 13-week-old AR-97Q mice during the dissection for immunohistochemistry and immunoblotting. We centrifuged the blood at $1000 \times g$ for 15 min and obtained the serum. The serum CK, AST, ALT, and BUN levels in the A419259- and vehicle-treated AR-97Q mice were measured using the method recommended by the Japanese Society of Clinical Chemistry (JSCC) at Mitsubishi Chemical Medience Corporation (Tokyo, Japan). The WBC, Hb and Plt levels in whole blood were measured using similar methods.

**Behavioral analyses.** All behavioral tests were performed weekly, and the data were analyzed prospectively. The animals' rotarod performance was assessed weekly using an Economex Rotarod (Ugo Basile, Comerio, Italy)[67]. A Grip Strength Meter (MK-380M, Muromachikikai, Tokyo, Japan) was used to measure the grip strength of the mice[68]. We performed three trials per week and recorded the best grip strength exhibited by each mouse.

**Immunoblotting.** Mice were deeply anesthetized, and the entire spinal cord and skeletal muscles were dissected and snap-frozen in powdered $CO_2$ in acetone. For the analysis of phosphorylated proteins, we dissected the mice 3 h after the last intraperitoneal injection at the age of 9 weeks. Tissues were homogenized in buffer containing 50 mM Tris-HCl (pH 8.0), 150 mM NaCl, 1% Nonidet P-40, 0.5% deoxycholate, 0.1% SDS, and 1 mM 2-mercaptoethanol with the Halt Protease and Phosphatase Inhibitor Cocktail (Thermo Scientific) and centrifuged at $2500 \times g$ for 15 min. Cultured cells were lysed in the same buffer after treatment. We separated equal amounts of protein on 5–20% SDS-PAGE gels (Wako, Osaka, Japan) and transferred them to Hybond-P membranes (GE Healthcare, Piscataway, NJ, USA). The following primary antibodies and dilutions were used: p-Src family (Tyr416) (#6943, 1:1000; Cell Signaling Technology), Src (#2108, 1:1000; Cell Signaling Technology), p-Stat3 (Tyr705) (#9145, 1:1000; Cell Signaling Technology), Stat3 (#4904, 1:1000; Cell Signaling Technology), p-p38MAPK (Thr180/Try182) (#9211, 1:1000; Cell Signaling Technology), p38MAPK (#9212, 1:1000; Cell Signaling Technology), AR (#5153, 1:2000; Cell Signaling Technology), p-Akt (Ser473) (#9271, 1:1000; Cell Signaling Technology), Akt (#9272, 1:1000; Cell Signaling Technology), p-p130Cas (Tyr410) (#4011, 1:1000; Cell Signaling Technology), p130Cas (#13846, 1:1000; Cell Signaling Technology), p-p44/42 MAPK (Erk1/2) (Thr202/Tyr204) (#4370, 1:1000; Cell Signaling Technology), p44/42 MAPK (Erk1/2) (#4695, 1:2000; Cell Signaling Technology), p-SAPK/JNK (Thr183/Tyr185) (#4668, 1:1000; Cell Signaling Technology), SAPK/JNK (#9252, 1:2000; Cell Signaling Technology), p-IκBα (Ser32) (sc-8404, 1:1000; Santa Cruz), IκBα (#4814, 1:1000; Cell Signaling Technology) cleaved caspase3 (Asp175) (#9661, 1:1000; Cell Signaling Technology) and Csk (#4980, 1:2000; Cell Signaling Technology). Primary antibodies bound to the proteins were probed with a 1:5000 dilution of horseradish peroxidase-conjugated secondary antibodies, and the bands were detected using an immunoreaction enhancing solution (Can Get Signal; Toyobo, Osaka, Japan) and enhanced chemiluminescence (ECL Prime; GE Healthcare). Chemiluminescence signals were digitized using a LAS-3000 imaging system (Fujifilm, Tokyo, Japan). The signal intensities of independent blots were quantified using IMAGE GAUGE software version 4.22 (Fuji) and are presented in arbitrary units. Membranes were re-probed with an anti-GAPDH antibody (MAB374, 1:5000; Santa Cruz Biotechnology) for normalization. Biopsy specimens of the skeletal muscles were obtained from patients with SBMA (32- and 50-year-old males) and control subjects (41-, 50-, 51-, and 52-year-old males) whose skeletal muscles are histologically normal. The collection and use of human tissues in this study were approved by the Ethics Review Committee of Nagoya University Graduate School of Medicine. Uncropped images are shown in the Source data file[69].

**Histology and immunohistochemistry.** Mouse tissues were dissected, post-fixed with 10% phosphate-buffered formalin and processed for paraffin embedding. Six-micron-thick sections were prepared from paraffin-embedded tissues. The sections designated to be stained with the anti-polyglutamine antibody (1C2) were treated with formic acid for 5 min at room temperature. The sections designated to be incubated with the anti-p-Src, ChAT, GFAP and p-p130Cas antibodies were boiled in 10 mM citrate buffer for 15 min. Primary antibodies bound to proteins were incubated with a secondary antibody labeled with a polymer as part of the Envision + system containing horseradish peroxidase (Dako Cytomation, Gostrup, Denmark). The following primary antibodies and dilutions were used to stain mouse tissues: p-Src (ab47411, 1:1000; Abcam), polyglutamine (MAB1574, 1:20,000; Millipore), ChAT (AB144P, 1:2000; Millipore), Neurofilament heavy chain (801701, 1:3000, Biolegend, San Diego, CA, USA), GFAP (#2301–1, 1:1000; Epitomics) and p-p130Cas (Tyr410) (PA5-37775, 1:20,000/1:6000; Invitrogen). Images of immunohistochemically stained sections were photographed using an optical microscope (BX51, Olympus, Tokyo, Japan). Immunoreactivity and cell size were analyzed using ImageJ software (NIH, Bethesda, MD). The means ± s.e.m. of the obtained values are presented in arbitrary units. For HE staining, 6-μm-thick cryostat sections of the gastrocnemius muscles were air-dried and stained. Autopsy specimens of the lumbar spinal cord or skeletal muscles were obtained from patients with genetically confirmed SBMA and control subjects: subjects with chronic inflammatory demyelinating polyneuropathy and myelodysplastic syndromes were used as controls for the spinal cord samples, and subjects with dementia presenting with Lewy bodies were used as controls for the skeletal muscle samples. The collection and use of human tissues in this study were approved by the Ethics Committee of Nagoya University Graduate School of Medicine. Uncropped images are shown in the Source data file[69].

**Quantitative analysis of immunohistochemistry.** At least 50 consecutive 6-μm-thick axial sections of the thoracic spinal cord and skeletal muscle were prepared, and every fifth section was immunostained with the 1C2 antibody to assess the number of positive cells. The numbers of 1C2-positive cells in all of the neurons of the anterior horn in 10 axial sections from the thoracic spinal cord from each group of mice ($n = 3$) were counted under a light microscope (Bx51). We defined a cell as a motor neuron when it was present within the anterior horn and showed an obvious nucleolus in each 6-μm-thick section. The numbers of 1C2-positive cells in skeletal muscles were calculated from more than 500 fibers in randomly selected areas of the 10 axial sections. We analyzed every fifth section of 25 consecutive 6-μm-thick axial sections from the thoracic spinal cord using ImageJ software (NIH) to quantify the size of the motor neurons. To calculate the cell size of the skeletal

muscles, More than 500 HE-stained fibers in randomly selected areas were examined using ImageJ software (NIH) to calculate the cell size in skeletal muscles.

**Immunoprecipitation**. We performed immunoprecipitation experiments using 300 μg of protein lysate from culture cells, 10 μL of the Src-specific antibody (ab16885) and a Dynabeads Protein G immunoprecipitation kit (Invitrogen), according to the manufacturer's instructions.

**Luciferase reporter assays**. To determine AR-dependent transcriptional activity, we transfected ARE-firefly luciferase and CMV-Renilla luciferase to NSC34 cells or C2C12 cells stably expressing AR-24Q or AR-97Q using Cignal Androgen Receptor Reporter (luc) Kit (Qiagen). Following 24 h of transfection, cells were treated with DHT or ethanol for 24 h. Firefly and Renilla luciferase substrates from Dual-Glo Luciferase Assay system (Promega, Wisconsin, USA) were added and luciferase activity was measured using a plate reader (2300 EnSpire™).

**Measurement of serum concentration of A419259**. The concentration of A419259 in the serum of AR-97Q mice were measured at Kola-Gen Pharma Co., Ltd. The serum was deproteinized by acetonitrile and examined with high performance liquid chromatography with ultraviolent detection (HPLC/UV) ($\lambda_1 = 300$ nm, $\lambda_2 = 320$ nm). The stability of A419259 in the serum of mice over time was evaluated. Calibration curves were linear (coefficient of determination = 0.9981) over a concentration range of 0.1–10 μg/mL.

**Site-directed mutagenesis**. The mutant construct that has deletions in 1345 to 1386 of AR-97Q was generated by site-directed mutagenesis using the KOD Plus mutagenesis kit (Toyobo) with the primers listed in Supplementary Table 1.

**Quantitative RT-PCR**. Total RNA was extracted from the cells using the RNeasy Mini Kit (Qiagen) and from NSC34 and C2C12 cells stably expressing AR-24Q or AR-97Q using TRIzol (Invitrogen). The extracted RNA was then reverse-transcribed into first-strand cDNA using the ReverTra Ace qPCR RT Kit (Toyobo). RT-PCR was performed using KOD SYBR qPCR kit (Toyobo) and the amplified products were detected with the iCycler system (Bio-rad Laboratories,Hercules, CA, USA). The reaction conditions were as follows: 98.0 °C for 2 min, 40 cycles of 10 s at 98.0 °C, 10 s at 60.0 °C and 30 s at 68.0 °C. The expression level of the internal control, *Gapdh*, was simultaneously quantified. The primers were listed in Supplementaty Table 1.

**Statistical analysis**. We analyzed the data using the unpaired two-sided *t*-test for comparisons of two groups and analysis of variance (ANOVA) with Dunnett's test for multiple comparisons in IBM SPSS Statistics 24 (Chicago, IL, USA). The survival rate was analyzed using the Kaplan–Meier and log-rank tests in IBM SPSS Statistics 24. We considered $p$ values of ≤0.05 to indicate statistical significance. The $p$ values for Bio-Plex phosphorylation protein arrays were adjusted for false discovery rate (FDR) based on the Benjamini and Hochberg method to correct for multiple testing. The adjustment was performed in each of the spinal cord and skeletal muscle. An FDR value of <0.10 was considered to be statistically significant. The correction for the multiple testing was conducted using SAS 9.4 (SAS Institute, Inc.).

**Study approval**. The collection of biopsied and autopsied human tissues and their use in this study were approved by the Ethics Review Committee of Nagoya University Graduate School of Medicine (No. 902–3), and written informed consent for the use of the specimens was obtained from the patients or patients' next of kin. Experimental procedures involving human subjects were conducted in accordance with the Declaration of Helsinki, the Ethics Guidelines for Human Genome/Gene Analysis Research and the Ethical Guidelines for Medical and Health Research Involving Human Subjects endorsed by the Japanese government. All animal experiments were performed in accordance with the National Institutes of Health Guide for the Care and Use of Laboratory Animals and with the approval of the Nagoya University Animal Experiment Committee (No. 25087).

**Reporting summary**. Further information on research design is available in the Nature Research Reporting Summary linked to this article.

## Data availability

A reporting summary for this Article is available as a Supplementary Information file. The source data underlying Figs. 1a–j, 2a–m, 3a–l, 4a–d, g–j, 5a–h, 6a–i, 7a–g, 8a–f and Supplementary Figs. 1a, b, 2a–d, 3a–f, 4a–c, 5a–g, 6a, b, 7a–d, 8a–d, 9a–d, 10a, b, 11a, b, 12a–d, 13a–c, 14a–c, 15a–d, 16a–d, 17a–c, 18a–d, 19a–l are provided as a Source Data file. The Source data has been deposited in Mendeley (https://doi.org/10.17632/sz4v5tcdyv.2)[69]. All data are available from the corresponding author upon reasonable request.

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

## Acknowledgements
This work was funded by Grants-in-Aid (KAKENHI) from the Ministry of Education, Culture, Sports, Science, and Technology of Japan (Nos. 26293206, 16K15480, 17H04195, 17J40128, and 18K15361), grants from the Japan Agency for Medical Research and Development (AMED) (No. 17ek0109221h0001 and 18ek0109221h0002), a grant from the Naito Foundation, a grant from the Hori Sciences and Arts Foundation, Nagoya University Hospital Funding for Clinical Research and a grant from "The Acceleration Program for Intractable Diseases Research utilizing Disease-Specific iPS cells" to H.O. from AMED (No. 19bm0804003h0003). No other agencies provided funding, and the investigators had sole discretion over the study design; collection, analysis, and interpretation of data; writing of the report; and the decision to submit it for publication.

## Author contributions
Project planning was performed by M.I., K.S., and M.K.; experiments were performed by M.I., K.S., N.K., H.N., G.T., Y.T., S.N., A.M., K.O., Y.O., Y.T-O., S.S., M.M., H.O., H.A., G.S., and M.K.; and data were analyzed by M.I., K.S., M.N., and M.K. The first draft of the manuscript was prepared by M.I. and M.K.; the manuscript layout was designed by K.S. and M.K.

## Additional information

**Competing interests:** The authors declare no competing interests.

