## [Peer Review File · Nature Communications]

Reviewers' comments:

Reviewer #1 (Remarks to the Author):

Iida et al. reported that an SBMA model and patients show increased levels of phosphorylated Src via p130Cas, and that Src kinase inhibition rescued phenotypes of SBMA. The findings are interesting in terms of neuromuscular degeneration and the manuscript is well organized, but there are some concerns to be resolved, as below.

Major points:

1. In Figure 1a, the level of p-Src in muscle of 13-wk SBMA model, mimicking advanced stage, is lower than that of control mice. However, autopsy cases of SBMA patients showed increased p-Src staining in muscle. These facts appear to be contradictory. From what stage of disease can the authors see a rise in p-Src by using patient samples and mouse models? It is important to clarify this point in the consideration of Src inhibitor as a future treatment for SBMA.
2. The title of the manuscript is not appropriate, since kinase inhibitors have multi-targets. The authors found the increase in phosphorylation of c-src using target specific antibody, but they did not conduct selective inhibition of c-src. The authors should add knock-down experiments.
3. The authors should clearly describe why they chose A41952 for in vitro and in vivo assays among the many Src inhibitors.
4. It is unclear how the authors determined the effective dose of A41952 both in vitro and in vivo. The authors should describe the experimental basis. Furthermore, the authors should analyze the dose-dependent effectiveness of A41952 at least in an in vitro assay.

Minor points:

1. In the introduction section, a lot of redundant and unnecessary information unrelated to the results will cause confusion for the readership. The authors should describe why the kinase pathway, including Src, was important in the pathology of SBMA, and why they decided to investigate kinase pathways comprehensively.
2. In Figure 1a, the authors conducted more than one hundred multiple comparisons, and they should have paid attention to type I errors of null hypothesis by adopting statistical correction, such as FDR-controlling procedures or Bonferroni correction.

3. In Figure 2, spell out DHT or add an abbreviation list.

4. The authors added 200 nM A41952 and showed improvement in p-Src and cellular phenotypes. 200 nM is a relatively high concentration, compared with IC50 of A41952, maybe around 9 nM. In some cases of kinase inhibitors, a high concentration can be harmful. The authors should clarify the dose-dependent cellular phenotypes in both 24Q- and 97Q cell lines to clarify whether direct cellular toxicity of A41952 has an effect or not.

5. In the experiments using SBMA model mice, the authors administered 0.5 mg/kg/day A41952 once every three days, but is this method enough to maintain optimal concentration of A41952 in plasma or CNS?

6. In Figure 6, the concentration of A41952 may be missing.

7. The authors should discuss the possibility of off-target effects of A41952 on SBMA phenotype improvement.

Reviewer #2 (Remarks to the Author):

The good points are: They Try to understand how polyQ of AR can influence the progress of SBMA, and provide a new agent as potential therapeutic option. They also suggest that a Src kinase inhibitor may ameliorate the behavioral changes and the histopathological phenotypes of the spinal cord and skeletal muscle in SBMA mice.

However, some weak points existed:

- 1) Do not explain AR as transcription factor how to regulate phosphorylation of Src.
- 2) Do not prove that alter the downstream p130CAS can reverse AR 97Q effect
- 3) The Neuron size different do not show convince data
- 4) No data for The interrupt assay for p130CAS
- 5) Fail to show how to specific target Src pathway but not influence other biological function
- 6) Some Figures have inconsistent control bands.
- 7) The author shall try which inhibitors of the SRC family have better effects, and see from there it may have benefit to the SMBA patients.

Other weakness are:

Introduction: Why OE rat AR in mice (typo??), and there is no statement for the polyglutamine-expanded AR mutant can induce neuromuscular degeneration.

Discussion

No explanation why src had no effect on the steady-state levels or aggregation of the pathogenic AR protein.

Figures:

Fig. 1: The Gapdh bands of Fig.1(B-C) are inconsistent. And in Fig. 1A, the patients should be age-sex paired patients, thus to eliminate selective bias.

Fig. 2. 2i the input very minor and control src bands are not even, and Fig. 2 should normalize the GAPDH of Fig. 1A, to see whether DHT could increase AR level. In addition, they can examine the AR activity changes after mutant AR in the presence DHT.

Fig. 4. In Fig 4h, the GAPDH band in muscle is not well. Fig 4C should be re-quantitated, the p value may be <0.01 . and Fig 4J should be re-quantitated, the p value may be less than 0.01.

Fig. 5. Fig 5H should be re-quantitated, the p value may be less than 0.01. and The chat and neuron size is not clear, shall provide more convincing data

Fig. 6. Should provide how p130CAS change phenotype and the reverse assay

NCOMMS-18-25623-T

Src inhibition attenuates polyglutamine-mediated neuromuscular degeneration in spinal and bulbar muscular atrophy

Point-by-point responses to the Reviewer's comments

Reviewer #1

Iida et al. reported that an SBMA model and patients show increased levels of phosphorylated Src via p130Cas, and that Src kinase inhibition rescued phenotypes of SBMA. The findings are interesting in terms of neuromuscular degeneration and the manuscript is well organized, but there are some concerns to be resolved, as below.

Major points:

Comment 1: In Figure 1a, the level of p-Src in muscle of 13-wk SBMA model, mimicking advanced stage, is lower than that of control mice. However, autopsy cases of SBMA patients showed increased p-Src staining in muscle. These facts appear to be contradictory. From what stage of disease can the authors see a rise in p-Src by using patient samples and mouse models? It is important to clarify this point in the consideration of Src inhibitor as a future treatment for SBMA.

Response 1: We thank the Reviewer for positively evaluating our manuscript. As the Reviewer pointed, Src phosphorylation in skeletal muscles is up-regulated in autopsied skeletal muscle of patients as well as in muscles of the mutant AR transgenic (AR-97Q) mice at their pre-onset to early stage of disease. However, Src phosphorylation is attenuated in skeletal muscles at the advanced stage of AR-97Q mice. To respond to the Reviewer's question, we compared the level of phosphorylated Src in the biopsied skeletal muscles of patients with SBMA and those of control subjects with immunoblotting. Biopsy was performed within 12 months and 7 years from the onset of weakness in SBMA patients, who were genetically diagnosed as SBMA later on. Phosphorylated Src was up-regulated in skeletal muscles of patients with SBMA compared with those of control subjects, indicating that Src phosphorylation is elevated in the skeletal muscle of SBMA patients at an early stage, in agreement with findings in the mouse model of SBMA (**Line 1-8, Page 6**). Compared with human pathology, muscular degeneration is markedly accelerated in AR-97Q mice due to a high expression level of the mutant AR which is controlled by a potent chicken beta-actin promoter (Katsuno et al., Neuron 2002). This leads to a massive myopathic change at the advanced stage of the model mice. By contrast, in patients with SBMA, nuclear and cytoplasmic accumulation of AR are virtually absent in the skeletal muscle, which shows less myogenic changes compared with AR-97Q mice (Adachi et al., Brain 2006). We thus consider that the difference in the severity of myopathic pathology between mice and patients underlies the differential pattern of Src phosphorylation in the skeletal muscle at advanced stage.

Comment 2: The title of the manuscript is not appropriate, since kinase inhibitors have multi-targets.

The authors found the increase in phosphorylation of c-src using target specific antibody, but they did not conduct selective inhibition of c-src. The authors should add knock-down experiments.

Response 2: We agreed with the Reviewer's opinion and thus added Src knock-down experiments using cellular models of SBMA. Selective inhibition of Src with siRNA improved the viability of cellular models of SBMA by reducing the toxicity of the mutant AR (**Line 8-10, Page 7; Supplementary Fig. 4**). We also performed additional experiments regarding the possible off-targets of the Src inhibitor we used, as described in our responses to Minor Comment No. 7 from Reviewer #1 and Comment No. 5 from Reviewer #2.

Comment 3: The authors should clearly describe why they chose A41952 for in vitro and in vivo assays among the many Src inhibitors.

Response 3: As the Reviewer suggested, we had administered SKI-1 and PP2 intraperitoneally to the mouse model of SBMA. However, they did not diminish Src phosphorylation in the spinal cord or skeletal muscle of AR-97Q mice (**Line 28-31, Page 9; Supplementary Fig. 12**). We thus chose A419259 for in vitro and in vivo assays, as this compound suppresses Src phosphorylation in a dose-dependent manner in both spinal cord and skeletal muscle of SBMA mice (**Supplementary Fig. 5a-c**). Limited bioavailability may underlie the ineffectiveness of SKI-1 and PP2 in vivo, given that these compounds mitigate the mutant AR toxicity in cellular models (**Supplementary Fig. 3**).

Comment 4: It is unclear how the authors determined the effective dose of A41952 both in vitro and in vivo. The authors should describe the experimental basis. Furthermore, the authors should analyze the dose-dependent effectiveness of A41952 at least in an in vitro assay.

Response 4: In vitro, we determined the effective dose of A41952 based on the dose-dependent effects of this compound on the viability and toxicity of the cellular models of SBMA (**Line 5, Page 7; Supplementary Fig. 2b, d**). In vivo assay, we first investigated the optimal concentration of A419259 to suppress Src phosphorylation in the spinal cord and skeletal muscle of AR-97Q mice. To this end, we administered 0.25, 0.5 or 1mg/kg/day of A419259 to AR-97Q mice once in three days for three times and anatomized them 2 hours after the last administration to analyze the level of phosphorylated Src in tissues. The results showed that the effects of A419259 on Src phosphorylation was stronger at 0.5 or 1mg/kg/day compared with 0.25 mg/kg/day both in the spinal cord and skeletal muscle of AR-97Q mice. As the effect was similar between 0.5 and 1mg/kg/day, we chose 0.5 mg/kg/day as the optimal dose (**Line 20-28, Page 7; Supplementary Fig. 5b, c**). As for the frequency of administering A419259 to AR-97Q mice, we investigated the duration of Src inhibition with A419259 at 0.5mg/kg/day by administering the compound to mice once in three days for three times and analyzing the Src phosphorylation levels in tissues one, three or four days after the last injection. The phosphorylated Src was down-regulated both in the spinal cord and skeletal muscle until three days after the last injection, but tended to elevate four days after the last administration (**Supplementary Fig. 5e-g**). Based on these findings, we determined the dose and frequency of A419259 to administer

to the mice as 0.5 mg/kg/day of A419259 once in three days (**Line 3-9, Page 8**). We also evaluated the effects of A419259 at 0.25mg/kg/day on the phenotype of AR-97Q mice, but this dose did not improve motor function or survival of SBMA mice (**Supplementary Fig. 7**).

Minor points:

Comment 1: In the introduction section, a lot of redundant and unnecessary information unrelated to the results will cause confusion for the readership. The authors should describe why the kinase pathway, including Src, was important in the pathology of SBMA, and why they decided to investigate kinase pathways comprehensively.

Response 1: In response to the Reviewer's comment, we re-organized the Introduction section and added the sentences explaining why we focused on the kinase pathways as follows: "Protein phosphorylation plays a major role in regulating protein functions in both neural and non-neural tissues. For instance, phosphorylation induces a conformational alteration in enzymes and receptors controlling their activity, regulates the transcription factor activities either positively or negatively and modifies the process of signal transduction. These crucial functions of protein phosphorylation led us to investigate the role of kinase activations in the pathogenesis of neurodegeneration in SBMA." (**Line 9-16, Page 4**).

Comment 2: In Figure 1a, the authors conducted more than one hundred multiple comparisons, and they should have paid attention to type I errors of null hypothesis by adopting statistical correction, such as FDR-controlling procedures or Bonferroni correction.

Response 2: According to the Reviewer's suggestion, we consulted with a biostatistician (M.N.) and conducted FDR-controlling procedures among the samples of spinal cords and skeletal muscles. We considered that FDR values of less than 0.1 indicate statistical significance. We mentioned this in the legend of **Fig. 1** and the Method section (**Line 31-36, Page 20**). Each FDR value was shown in the separate Source Data file.

Comment 3: In Figure 2, spell out DHT or add an abbreviation list.

Response 3: We spelled out DHT in the legend of **Fig. 2**.

Comment 4: The authors added 200 nM A41952 and showed improvement in p-Src and cellular phenotypes. 200 nM is a relatively high concentration, compared with IC₅₀ of A41952, maybe around 9 nM. In some cases of kinase inhibitors, a high concentration can be harmful. The authors should clarify the dose-dependent cellular phenotypes in both 24Q- and 97Q cell lines to clarify whether direct cellular toxicity of A41952 has an effect or not.

Response 4: In response to the Reviewer's comment, we examined the dose-dependent effect of A419259 on the cell viability and toxicity in cells stably expressing AR-24Q or AR-97Q (**Supplementary Fig. 2**). A419259 at 20 and 200 nM up-regulated the viability of AR-24Q cells to a

lesser extent than AR-97Q cells and had no effect on LDH release of AR-24Q cells, suggesting that the direct cellular toxicity of A419259 is absent at the concentrations we utilized.

Comment 5: In the experiments using SBMA model mice, the authors administered 0.5 mg/kg/day A41952 once every three days, but is this method enough to maintain optimal concentration of A41952 in plasma or CNS?

Response 5: We measured the concentration of A419259 in the serum of AR-97Q mice 10 minutes after the injection of each dose of A419259. The concentrations of A419259 in the serum of the mice treated with 0.25mg/kg/day A419259 were approximately half of those in the mice treated with 0.5mg/kg/day A419259, whereas the serum concentrations in the mice treated with 1.0 mg/kg/day A419259 were similar to those in the mice treated with 0.5mg/kg/day of A419259, suggesting that the effect of Src kinase inhibition of A419259 is dose-dependent till 0.5mg/kg/day (**Supplementary Fig. 5d**). These results support the findings in immunoblots: the expression levels of phosphorylated Src in the spinal cord and skeletal muscle of the mice treated with 0.5mg/kg/day of A419259 are similar to those in the mice treated with 1.0mg/kg/day (**Supplementary Fig. 5b and c**).” (**Line 28, Page 7 to Line 2, Page 8**).

Comment 6: In Figure 6, the concentration of A41952 may be missing.

Response 6: We added the concentration of A419259 in the legend of **Fig. 6**.

Comment 7: The authors should discuss the possibility of off-target effects of A41952 on SBMA phenotype improvement.

Response 7: In response to the Reviewer's comment, we provided the effects of A419259 on the phosphorylation levels of other molecules in the spinal cord and skeletal muscle of AR-97Q mice. As shown in **Supplementary Fig. 6**, administration of A419259 at 0.5mg/kg/day did not alter the phosphorylation of p38MAPK, JNK, I κ B α or p44/42MAPK. We investigated the phosphorylation levels of these molecules as they play important roles in cellular survival and function and some of them also modify the pathogenesis of SBMA (Minamiyama et al, Nat Med 2012; Iida et al., Hum Mol genet 2015) (**Line 10-17, Page 8; Supplementary Fig. 6**).

Reviewer #2

The good points are: They Try to understand how polyQ of AR can influence the progress of SBMA, and provide a new agent as potential therapeutic option. They also suggest that a Src kinase inhibitor may ameliorate the behavioral changes and the histopathological phenotypes of the spinal cord and skeletal muscle in SBMA mice.

However, some weak points existed:

Comment 1: Do not explain AR as transcription factor how to regulate phosphorylation of Src.

Response 1: Over-expression of AR has virtually no effect on the protein levels of Src both in vivo and in vitro (**Fig. 1b and 2c**), suggesting that AR regulates Src activity at a post-translational phase. In support of this view, ligand-induced activation of AR is shown to upregulate the autophosphorylation of Src, whereas culture of a prostate tumor cell line under androgen ablation conditions dramatically attenuates Src autophosphorylation (Migliaccio et al., *Embo J* 2000). In addition, androgen induces the assembly of a ternary complex comprising AR and Src, thereby triggering stronger stimulation of Src activity. Furthermore, inhibition of AR expression using an siRNA decreased Src activation but not total Src expression in a cellular model of prostate cancer, indicating that AR directly regulates Src activation but not its expression (Zarif et al., *Oncotarget* 2015). Similarly, inhibition of AR expression in other prostate tumor cell lines using siRNAs suppressed Src activation in response to R1881, a synthetic androgen methyltrienolone (**Line 16-26, Page 12**).

Comment 2: Do not prove that alter the downstream p130CAS can reverse AR 97Q effect

Response 2: We now showed that the suppression of p130Cas using siRNA increased cell viability and suppressed cell death of NSC34 and C2C12 cells stably expressing AR-97Q (**Fig. 8a-c**). On the contrary, overexpression of p130Cas decreased cell viability and increased cell death of AR-97Q cells (**Fig. 8d-f**). Furthermore, we performed the experiments to investigate the downstream molecules of p130Cas and found that the down-regulation of p130Cas decreases cleaved caspase 3 levels and overexpression of p130Cas up-regulates cleaved caspase 3, suggesting that p130Cas induces cellular damage by activating apoptotic pathway. We explained these findings in the Results section (**Line 4-9, Page 11**).

Comment 3: The Neuron size different do not show convince data

Response 3: To answer the Reviewer's comment, we added the data of immunoblots for ChAT of the spinal cords of wild-type and AR-97Q mice treated with or without A419259 (**Supplementary Fig. 11**). We added the following sentences: "Immunoblots revealed that protein levels of ChAT in the spinal cord are decreased in vehicle-treated AR-97Q mice compared with wild-type mice, but increased by the administration of A419259 (**Supplementary Fig. 11**)." **(Line 23-25, Page 9)**.

Comment 4: No data for The interrupt assay for p130CAS

Response 4: In response to the Reviewer's suggestion, we additionally examined the effect of bosutinib, a p130Cas inhibitor, on the toxicity of polyglutamine-expanded AR in the cellular models of SBMA. We described the results as follows: "Moreover, bosutinib, a p130Cas inhibitor, increased cell viability and the cell number and attenuated LDH release in the cellular models of SBMA (**Supplementary Fig. 13**)." **(Line 4-9, Page 11)**.

Comment 5: Fail to show how to specific target Src pathway but not influence other biological function

Response 5: In response to the Reviewer's comment, we provided the effects of A419259 on the

phosphorylation levels of other molecules in the spinal cord and skeletal muscle of AR-97Q mice. As shown in **Supplementary Fig. 6**, administration of A419259 at 0.5mg/kg/day did not alter the phosphorylation of p38MAPK, JNK, I κ B α or p44/42MAPK. We investigated the phosphorylation levels of these molecules as they play important roles in cellular survival and function and some of them also modify the pathogenesis of SBMA (Minamiyama et al, Nat Med 2012; Iida et al., Hum Mol genet 2015) (**Line 10-17, Page 8; Supplementary Fig. 6**).

Comment 6: Some Figures have inconsistent control bands.

Response 6: As for **Fig.1b** and **c**, we confirmed that the signal intensity of Gapdh bands is similar between wild-type and AR-97Q in the triplicated immunoblots as shown in the figure for Reviewer shown below. In **Fig. 2a**, the intensities of Gapdh bands were not equal among four samples so that we re-performed immunoblotting of human AR and Gapdh to load the equal amount of Gapdh.

Fig. 1b

Fig. 1c

Comment 7: The author shall try which inhibitors of the SRC family have better effects, and see from there it may have benefit to the SMBA patients.

Response 7: As the Reviewer suggested, we had administered SKI-1 and PP2 intraperitoneally to the mouse model of SBMA. However, they did not diminish Src phosphorylation in the spinal cord or skeletal muscle of AR-97Q mice (**Line 28-31, Page 9; Supplementary Fig. 12**). We thus chose A419259 for in vitro and in vivo assays, as this compound suppresses Src phosphorylation in a dose-dependent manner in both spinal cord and skeletal muscle of SBMA mice (**Supplementary Fig. 5a-c**). Limited bioavailability may underlie the ineffectiveness of SKI-1 and PP2 in vivo, given that these compounds mitigate the mutant AR toxicity in cellular models (**Supplementary Fig. 3**).

Other weakness are:

Introduction

Comment: Why OE rat AR in mice (typo??), and there is no statement for the polyglutamine-expanded AR mutant can induce neuromuscular degeneration.

Response: Overexpression of wild-type rat AR in muscle is shown to induce motor neuron damage in

mice (Monks et al., Proc Natl Acad Sci USA 2007). As shown in other polyglutamine-mediated disorders, hyper-activation of native function of AR protein, via interaction with coregulators, underlies the toxicity of the polyglutamine-expanded AR (Nedelsky et al., Neuron 2010; Orr, Curr Opin Genet Dev 2012). In addition, several lines of experimental evidence indicate the muscular pathology plays a primary role in SBMA. Muscle-specific excision of the mutant AR gene improves motor phenotype, neuropathology and survival in a BAC transgenic mouse model of SBMA (Cortes et al., Neuron 2014). Peripheral gene silencing of the mutant AR by antisense oligonucleotides also alleviates motor dysfunction and neurodegeneration in a knock-in mouse model of SBMA, while gene silencing in the central nervous system also results in attenuation of neuromuscular phenotype of a transgenic mouse harboring polyglutamine-expanded AR (Lieberman et al., Cell Rep 2015; Sahashi et al., Hum Mol Genet 2016). These findings provide a theoretical basis for the observation that overexpression of wild-type AR in skeletal muscle causes motor neuron damage, and underscore the importance of an altered neuron-muscle interplay in the pathogenesis of SBMA. We added these issues to Introduction (**Line 12-18 and Line 30-36, Page 3**).

Discussion

Comment: No explanation why src had no effect on the steady-state levels or aggregation of the pathogenic AR protein.

Response: AR is regulated by several post-translational modifications such as phosphorylation, acetylation and methylation (Beitel 2013). Phosphorylation at each site leads to an increase, no change or a decrease in AR transactivity (Yeh et al., PNAS 1999, Lin et al., PNAS 2001). Some phosphorylation sites are reported to improve the pathogenesis of SBMA and others are revealed to aggravate it. Akt phosphorylates the wild-type AR at Ser215 and Ser792 (Lin et al., PNAS 2001, Palazzolo et al., Neuron 2009). Phosphorylation of AR-65Q by Akt at these sites reduced androgen binding and transcriptional activation, decreased ligand-induced stabilization and nuclear translocation of AR (Palazzolo et al., Neuron 2009). In vivo, enhancement of IGF-1/Akt pathway rescues behavioral and histopathological abnormalities of AR-97Q mice. A mutation of polyglutamine-expanded AR in the FxxLF motif (F23A) to prevent the N/C interaction induces Serine16 phosphorylation in AR, resulting in improvement of motor function in another mouse model of SBMA (Zboray et al., Cell Rep 2015). On the contrary, phosphorylation at other sites of AR exacerbates the phenotype of SBMA. In cells expressing polyglutamine-expanded AR, phosphorylation of AR at Ser512 via the p44/42 MAPK pathway induces cell death by enhancing the function of caspase 3 to generate cytotoxic polyglutamine fragments (LaFevre-Bernt et al., J Biol Chem 2003). Cyclin-dependent kinase 2 (CDK2) phosphorylates polyglutamine AR at Ser96 and thereby increases stabilization and toxicity of mutant AR (Polanco et al., Sci Transl Med 2016). However, the effect of Src on these phosphorylation sites in AR has not been reported. We added these issues to the Discussion section (**Line 35, Page 12 to Line 5, Page 13**).

Figures:

Comment for Fig. 1: The Gapdh bands of Fig.1(B-C) are inconsistent. And in Fig. 1H, the patients should be age-sex paired patients, thus to eliminate selective bias.

Response: As for **Fig.1b** and **c**, we confirmed that the signal intensity of Gapdh bands is similar between wild-type and AR-97Q in the triplicated immunoblots as shown in the figure for Reviewer shown below. We also changed the immunostaining samples in **Fig.1h**: 74 and 57-year-old males for Control patients, and 77 and 52-year-old males for SBMA patients.

Fig. 1b

Fig. 1c

Comment for Fig. 2: 2i the input very minor and control src bands are not even, and Fig. 2 should normalize the GAPDH of Fig. 2A, to see whether DHT could increase AR level. In addition, they can examine the AR activity changes after mutant AR in the presence DHT.

Response: We revised **Fig.2j** by increasing the amount of input, so that Src bands became even. We re-performed immunoblotting of AR and GAPDH to normalize the expression level of GAPDH in **Fig. 2a**. We also performed quantitative densitometry analysis in **Fig. 2b**, and found that DHT increases AR expression levels in NSC34 and C2C12 cells stably expressing AR-24Q or AR-97Q. As for the transcriptional activity of AR, we conducted a dual luciferase reporter assay and showed that DHT treatment increases transactivity of AR in NSC34 and C2C12 cells, regardless the length of polyglutamine tract (**Supplementary Fig. 1**).

Comment for Fig. 4: In Fig 4h, the GAPDH band in muscle is not well. Fig 4C should be re-quantitated, the p value may be <0.01. and Fig 4J should be re-quantitated, the p value may be less than 0.01.

Response: We re-pasted full bands of GAPDH of muscle samples in **Fig.4h**. In response to the Reviewer's criticism, we confirmed the results of statistical analysis and corrected the expression of p values (**Fig. 4c and j**).

Comment for Fig. 5: Fig 5H should be re-quantitated, the p value may be less than 0.01. and The chat and neuron size is not clear, shall provide more convincing data

Response: The p value of **Fig. 5h** was less than 0.01 and we changed "<0.05" to "<0.01". We made

similar corrections in **Figs. 2-8 and Supplementary Figs. 3 and 17** as well. As we mentioned in the response for the Major Comment No. 3 of Reviewer #2, we added the data of immunoblots for ChAT of the spinal cords of wild-type and AR-97Q mice treated with or without A419259 (**Supplementary Fig. 11**). We added the following sentences: “Immunoblots revealed that protein levels of ChAT in the spinal cord are decreased in vehicle-treated AR-97Q mice compared with wild-type mice, but increased by the administration of A419259 (**Supplementary Fig. 11**).” (**Line 23-25, Page 9**).

Comment for Fig. 6: Should provide how p130CAS change phenotype and the reverse assay

Response: We have revealed the suppression of phosphorylated p130Cas using siRNA increased cell viability and the cell number and diminished LDH release in AR-97Q cells (**Fig. 8a-c**). On the contrary, overexpression of p130Cas decreased cell viability and the cell number and increased LDH release in AR-97Q cells (**Fig. 8d-f**). Following the Reviewer’s suggestion, we performed the experiments to reveal the effect of bosutinib, a p130Cas inhibitor, on cellular models of SBMA. We mentioned the results as follows: “Moreover, bosutinib, a p130Cas inhibitor, increased cell viability and attenuated LDH release in cellular models of SBMA (**Supplementary Fig. 13**).” (**Line 4-6, Page 11**). Furthermore, we investigated the downstream molecules to p130Cas and showed that the down-regulation of p130Cas decreased cleaved caspase 3 levels and that overexpression of p130Cas augments cleaved caspase 3, suggesting that activation of p130Cas activates apoptosis pathway. We explained these findings in the Results section (**Line 6-9, Page 11**).

Reviewers' comments:

Reviewer #1 (Remarks to the Author):

lida et al. revised their manuscript according to this reviewer's comments, and the manuscript is improved and well organized. However, the authors need additional revision to clarify the data significance, as below.

Major points:

1. The authors presented figures of original gel images from western blot analysis in the supplemental file. However, there are no labels for each of the samples. Also, some figures are not shown as full blots, eg., Fig.7c, Fig.8a, Fig.12a, Fig.12c in the supplemental file. The authors should present full blots and label sample names in the supplemental figures.

Reviewer #2 (Remarks to the Author):

Authors failed to answer well following critiques:

(1) Comment 1-Fig1 j result: pSrc increase as well as the internal control

(2) For review2 comment-1 cite the reference but do not explain the reason why only change phosphorylation level

(3) For review2 comment-3 immunoblots for ChAT is not appropriate indication for the neuron size.

Other weak points still existed in this revised manuscript:

(1) In the response to the reviewer's comment, the author tried to explain how AR regulate phosphorylation of Src. They did show some evidence that AR induced phosphorylation of Src. However, they still failed to provide mechanism evidence of how. This is also true that in overall this manuscript missed many detailed mechanism dissection.

(3) Some figures we can see The loading control band also increased as target gene, which make result not easy to judge

(4) The photo in fig 5 do not match the statistical significance; Fig. 5 c-d The representative figure of AR-97Q and quantification not match each other.

(5) In figure 8a, Two siRNAs are required to exclude the off-target effect

NCOMMS-18-25623-A

Src inhibition attenuates polyglutamine-mediated neuromuscular degeneration in spinal and bulbar muscular atrophy

Point-by-point responses to the Reviewer's comments

Reviewer #1

Iida et al. revised their manuscript according to this reviewer's comments, and the manuscript is improved and well organized. However, the authors need additional revision to clarify the data significance, as below.

Major points:

Comment 1: The authors presented figures of original gel images from western blot analysis in the supplemental file. However, there are no labels for each of the samples. Also, some figures are not shown as full blots, eg., Fig.7c, Fig.8a, Fig.12a, Fig.12c in the supplemental file. The authors should present full blots and label sample names in the supplemental figures.

Response 1: We thank the Reviewer for positively evaluating our manuscript. In response to the Reviewer's comment, we labeled each sample in the supplemental file. As for the figures pointed by the Reviewer (Fig.6d, 7c, 8a, Supplementary Fig.12a, 12c, 16a, 16c, 18a and 18c), we cut membranes before antibody probing to examine multiple proteins with different molecular weight on the same membrane and/or save the amount of antibodies to be used. We revised the original gel images of Fig. 6d, 8a, Supplementary Fig. 12a, 12c, 16a, 16c, 18a and 18c to clarify how we used membranes for multiple probing.

Reviewer #2

Major points:

Comment 2: Comment 1-Fig1 j result: pSrc increase as well as the internal control

Response 2: In response to the Reviewer's comment, we re-performed immunoblotting of phosphorylated and total Src and Gapdh so that we loaded the equal amount of Gapdh in **Fig. 1j**.

Comment 3: For review2 comment-1 cite the reference but do not explain the reason why only change phosphorylation level

Response 3: Thank you again for pointing this important issue. To answer the Reviewer's comment, we performed RT-PCR analysis to measure mRNA levels of *Src* in NSC34 and C2C12 cells stably expressing AR-24Q or AR-97Q. The results showed that the mRNA levels of *Src* in 97Q cells were not

different from those in 24Q cells, suggesting that AR up-regulates the activity of Src at a post-translational phase (**Line 20-23, Page 6; Fig. 2f and g**). As C-terminal Src kinase (Csk) is a major regulator of Src, which catalyzes the phosphorylation of Src at Tyr527 that suppresses the auto-phosphorylation of Src at Tyr416, we then investigated the protein and mRNA levels of Csk in NSC34 and C2C12 cells stably expressing AR-24Q or AR-97Q. The results demonstrated that Csk was up-regulated by AR-97Q, suggesting that the up-regulated Src phosphorylation at Tyr416 in SBMA was not mediated by Csk. Therefore we further investigated the mechanisms of Src activation in SBMA in light of the direct interaction of Src with AR, as several reports have clearly shown that the direct association of AR with Src triggers the auto-phosphorylation of Src (Migliaccio et al. *J Steroid Biochem Mol Biol*, 2002; Shupnik et al. *Oncogene*, 2004). The SH3 domain of Src interacts with 372-385 amino acid residues in human AR (Migliaccio et al. *Oncogene*, 2007), which correspond to the DNA sequence of 1345-1386 in the AR-97Q vector. We created the AR construct with a deletion of 1345-1386 (deleted AR-97Q) using site-directed mutagenesis procedures and confirmed that the deleted AR-97Q have a weaker association with Src, compared with full-length AR-97Q, by performing immunoprecipitation (**Supplementary Fig. 19e**). The phosphorylation levels of Src was greatly decreased in the NSC34 cells co-transfecting with the Src vector and the deleted AR-97Q vector, compared with those in the cells co-transfecting with the Src vector and full-length AR-97Q vector (**Supplementary Fig. 19f, g**). We next examined the impact of SH3 domain deletion on Src phosphorylation in the cellular model of SBMA. We first confirmed that Src lacking the SH3 domain (SH3 Δ Src) can still be activated by a Src activator MLR1023 (**Supplementary Fig. 19h**) and that the SH3 Δ Src have a weaker association with AR, compared with the full-length Src in the NSC34 cells stably expressing AR-97Q (**Supplementary Fig. 19i**). The phosphorylation level of SH3 Δ Src was diminished compared with the full-length Src, in the NSC34 cells stably expressing AR-97Q (**Supplementary Fig. 19j, k**). Together, these findings suggest that Src activation in SBMA arises from the protein-protein interaction between Src with AR, not from the function of AR as a transcription factor. We explained these issues in the Results section (**Line 32, Page 11- Line 2, Page 13**) and Discussion section (**Line 1-3, Page 15**).

Comment 4: For review2 comment-3 immunoblots for ChAT is not appropriate indication for the neuron size.

Response 4: In response to the Reviewer's comment, we additionally performed immunostaining for neurofilament heavy chain of the spinal cords from wild type and AR-97Q mice treated with or without A419259, instead of immunoblots for ChAT (**Supplementary Fig. 11**). Unphosphorylated neurofilament heavy chain, SMI32, is observed in neuronal soma and used for staining motor neurons in several articles (Chevalier-Larsen ES et al. *Dis Model Mech* 2012; Montoya G JV et al. *Exp Neurol* 2009). The immunostaining for this neurofilament revealed that the size of the motor neurons in vehicle-treated AR-97Q mice were decreased compared with wild-type mice, and that the size was restored by the treatment with A419259. We mentioned the experiment in the Result section (**Line**

Minor points:

Comment 5: In the response to the reviewer's comment, the author tried to explain how AR regulate phosphorylation of Src. They did show some evidence that AR induced phosphorylation of Src. However, they still failed to provide mechanism evidence of how. This is also true that in overall this manuscript missed many detailed mechanism dissection.

Response 5: Please refer to the description in our responses to Major Comment No. 3 from Reviewer #2.

Comment 6: Some figures we can see The loading control band also increased as target gene, which make result not easy to judge

Response 6: In response to the Reviewer's comment, we re-performed immunoblotting to equalize the amount of Gapdh of control bands (**Fig. 6g** and **Fig. 7c**).

Comment 7: The photo in fig 5 do not match the statistical significance; Fig. 5 c-d The representative figure of AR-97Q and quantification not match each other.

Response 7: We thank the Reviewer for pointing this issue. In response to the Reviewer's comment, we exchanged the image of the skeletal muscle of AR-97Q mice treated with A419259 in **Fig. 5c** so that it matches the result of quantification of 1C2 positive cells in **Fig. 5d**.

Comment 8: In figure 8a, Two siRNAs are required to exclude the off-target effect.

Response 8: To exclude the possibility of an off-target effect, we added experiments using another siRNA for p130Cas (siRNA-2) (**Supplementary Fig. 13**), the results of which demonstrated the similar findings as our original experiments using siRNA-1 for p130Cas (**Fig. 8a**).

Reviewers' comments:

Reviewer #2 (Remarks to the Author):

After we reviewing this revised manuscript, we still have 2 critiques.

1. Comment 4-5

If AR regulates Src activation independent of its transcriptional activity, how DHT works to promote p-Src. Authors need to answer with data

2. Comment 7

It is not appropriate that simply exchange the pictures to match the quantification results. The author has to provide all pictures as supplementary data.

NCOMMS-18-25623-B

Src inhibition attenuates polyglutamine-mediated neuromuscular degeneration in spinal and bulbar muscular atrophy

Point-by-point responses to the Reviewer's comments

Reviewer #2

Comment 1: Comment 4-5. If AR regulates Src activation independent of its transcriptional activity, how DHT works to promote p-Src. Authors need to answer with data.

Response 1: We thank the Reviewer for pointing this issue. In response to the Reviewer's comment, we investigated the effect of DHT on the interaction of Src with AR, as androgen is reported to induce the assembly of a ternary complex comprising AR and Src, thereby triggering stronger stimulation of Src activity (Migliaccio et al., *Embo j* 2000). To assess the interaction of Src with AR-97Q in the cells treated with or without DHT, we transfected NSC34 cells stably expressing AR-97Q with the full-length Src vector and treated the cells with ethanol or DHT. We then immunoprecipitated the cell lysate using the Src antibody and immunoblotted the proteins with the AR antibody. We detected the strong association of Src with AR-97Q in the cells treated with DHT, whereas Src and AR-97Q were hardly associated with each other without DHT treatment (**Supplementary Fig. 19I**). These findings suggested that DHT activates Src signaling pathway by promoting the firm interaction of AR with Src.

Comment 2: Comment 7. It is not appropriate that simply exchange the pictures to match the quantification results. The author has to provide all pictures as supplementary data.

Response 2: We entirely agree with the Reviewer's opinion. In response to the Reviewer's comment, we provided all the raw data which we used for quantifications as supplementary data (**Fig.5b, d, f, h and Supplementary Fig.11b**).

REVIEWERS' COMMENTS:

Reviewer #2 (Remarks to the Author):

All of the questions have been answered well. Now the manuscript can be accepted.